# LIKELIHOOD-BASED PERMUTATION INVARIANT LOSS FUNCTION FOR PROBABILITY DISTRIBUTIONS

## ABSTRACT

We propose a permutation-invariant loss function designed for the neural networks reconstructing a set of elements without considering the order within its vector representation. Unlike popular approaches for encoding and decoding a set, our work does not rely on a carefully engineered network topology nor by any additional sequential algorithm. The proposed method, Set Cross Entropy, has a natural information-theoretic interpretation and is related to the metrics defined for sets. We evaluate the proposed approach in two object reconstruction tasks and a rule learning task.

## 1 INTRODUCTION

Sets are fundamental mathematical objects which appear frequently in the real-world dataset. However, there are only a handful of studies on learning a set representation in the machine learning literature. In this study, we propose a new objective function called *Set Cross Entropy* (SCE) to address the permutation invariant set generation. SCE measures the cross entropy between two sets that consists of multiple elements, where each element is represented as a multi-dimensional probability distribution in $[0, 1] \subset \mathbb{R}$ (a closed set of reals between 0,1). SCE is invariant to the object permutation, therefore does not distinguish two vector representations of a set with the different ordering. The SCE is simple enough to fit in one line and can be naturally interpreted as a formulation of the log-likelihood maximization between two sets derived from a logical statement.

The SCE loss trains a neural network in a permutation-invariant manner, and the network learns to output a set. Importantly, this is *not* to say that the neural network *learns to represent* a function that is permutation-invariant with regard the input. The key difference in our approach is that we allow the network to output a vector representation of a set that may have a different ordering than the examples used during the training. In contrast, previous studies focus on learning a function that returns the same output for the different permutations of the input elements. Such scenarios assume that an output value at some index is matched against the target value at the same index. [1]

This characteristic is crucial in the tasks where the objects included in the supervised signals (training examples for the output) do not have any meaningful ordering. For example, in the logic rule learning tasks, a first-order logic horn clause does not care about the ordering inside the rule body since logical conjunctions are invariant to permutations, e.g. $\mathsf{father(c, f)} \leftarrow (\mathsf{parent(c, f)} \wedge \mathsf{male(f)})$ and $\mathsf{father(c, f)} \leftarrow (\mathsf{male(f)} \wedge \mathsf{parent(c, f)})$ are equivalent.

To apply our approach, no special engineering of the network topology is required other than the standard hyperparameter tuning. The only requirement is that the target output examples are the probability vectors in $[0, 1]^{N \times F}$, which is easily addressed by an appropriate feature engineering including autoencoders with softmax or sigmoid latent activation.

We demonstrate the effectiveness of our approach in two object-set reconstruction tasks and the supervised theory learning tasks that learn to perform the backward chaining of the horn clauses. In particular, we show that the SCE objective is superior to the training using the other set distance metrics, including Hausdorff and set average (Chamfer) distances.

---

[1] For example, the permutation-equivariant/invariant layers in Zaheer et al. (2017) are evaluated on the classification tasks, the regression tasks and the set expansion tasks. In the classification and the regression tasks, the network predicts a single value, which has no ordering. In the set expansion tasks, the network predicts the probability $p_i$ for each tag $i$ in the output vector. Thus, reordering the output does not make sense.

## 2 BACKGROUNDS AND RELATED WORK

### 2.1 LEARNING A SET REPRESENTATION

Previous studies try to discover the appropriate structure for the neural networks that can represent a set. Notable recent work includes permutation-equivariant / invariant layers that addresses the permutation in the input (Guttenberg et al., 2016; Ravanbakhsh et al., 2016; Zaheer et al., 2017).

Let $X$ be a vector representation of a set $\{x_1, \ldots, x_n\}$ and $\pi$ be an arbitrary permutation function for a sequence. A function $f(X)$ is *permutation invariant* when $\forall \pi; f(X) = f(\pi(X))$. Zaheer et al. (2017) showed that functions are permutation-invariant iff it can be decomposed into a form

$$f(X) = \rho \left( \sum_{x \in X} \phi(x) \right)$$

where $\rho, \phi$ are the appropriate mapping function.

However, as mentioned in the introduction, the aim of these layers is to learn the functions that are permutation-invariant with regard to the input permutation, and not to reconstruct a set in a permutation-invariant manner (i.e. ignoring the ordering). In other words, permutation-equivariant/invariant layers are only capable of encoding a set.

Probst (2018) recently proposed a method dubbed as "Set Autoencoder". It additionally learns a permutation matrix that is applied before the output so that the output matches the target. The target for the permutation matrix is generated by a Gale-Shapley greedy stable matching algorithm, which requires $O(n^2)$ runtime. The output is compared against the training example with a conventional loss function such as binary cross entropy or mean squared error, which requires the final output to have the same ordering as the target. Therefore, this work tries to learn the set as well as the ordering between the elements, which is conceptually different from learning to reconstruct a set while ignoring the ordering.

Another line of related work utilizes Sinkhorn iterations (Adams & Zemel, 2011; Santa Cruz et al., 2017; Mena et al., 2018) in order to directly learn the permutations. Again, these work assumes that the output is generated in a specific order (e.g. a sorting task), which does not align with the concept of the set reconstruction.

### 2.2 SET DISTANCE

Set distances / metrics are the binary functions that satisfy the metric axioms. They have been utilized for measuring the visual object matching or for feature selection (Huttenlocher et al., 1993; Dubuisson & Jain, 1994; Piramuthu, 1999). Note that, however, in this work, we use the informal usage of the terms "distance" or "metric" for any non-negative binary functions that may not satisfy the metric axioms.

There are several variants of set distances. Hausdorff distance between sets (Huttenlocher et al., 1993) is a function that satisfies the metric axiom. For two sets $X$ and $Y$, the directed Hausdorff distance with an element-wise distance $d(x, y)$ is defined as follows:

$$\mathcal{H}_{1d}(X, Y) = \max_{x \in X} \min_{y \in Y} d(x, y)$$

The element-wise distance $d$ is Euclidean distance or Hamming distance, for example, depending on the target domain.

Set average (pseudo) distance (Dubuisson & Jain, 1994, Eq.(6)), also known as Chamfer distance, is a modification of the original Hausdorff distance which aggregates the element-wise distances by summation. The directed version is defined as follows:

$$A_{1d}(X, Y) = \frac{1}{|X|} \sum_{x \in X} \min_{y \in Y} d(x, y).$$

Set average distance has been used for image matching, as well as to autoencode the 3D point clouds in the euclidean space for shape matching (Zhu et al., 2016).

## 3   SET CROSS ENTROPY

Inspired by the various set distances, we propose a straightforward formulation of likelihood maximization between two sets of probability distributions. In what follows, we define the cross entropy between two sets $X, Y \in [0, 1]^{N \times F}$, where $[0, 1]$ is a closed set of reals between 0 and 1.

Let $\mathcal{X} = \left\{ X^{(1)}, X^{(2)}, \ldots \right\}$ be the training dataset, and $Y$ be the output matrix of a neural network. Assume that each $X \in \mathcal{X}$ consists of $N$ elements where each element is represented by $F$ features, i.e. $X = \{x_1 \ldots x_N\}, x_i \in \mathbb{R}^F$. We further assume that $x_i \in [0, 1]^F$ by a suitable transformation, which can be done by the feature learning with sigmoid activation added to the latent layer. The set $X$ actually takes the vector representation, which essentially makes $X \in [0, 1]^{N \times F}$. In this paper, we focus on the binomial distribution. However, the proposed method naturally extends to the multinomial case.

For simplicity, we assume that the number of elements in the set $X$ and $Y$ is known and fixed to $N$. Therefore $Y$ is also a matrix in $[0, 1]^{N \times F}$. Furthermore, we assume that $X$ is preprocessed and contains no duplicated elements.

In practice, if $|X|$ varies across the dataset, it suffices to take $N^{\max} = \max_{X \in \mathcal{X}} |X|$, the largest number of elements in $X$ across the dataset $\mathcal{X}$, and add the dummy, distinct objects $d_0 \ldots d_{N^{\max}}$ to fill in the blanks. For example, when there are $N$ objects of $F$ features and we want to normalize the size of the set to $N'(> N)$, one way is to add an additional axis to the feature vector ($F + 1$ features) where the additional $F + 1$-th feature is 0 for the real data and 1 for the dummy data, and the additional $N' - N$ objects are generated in an arbitrary way (e.g. as a binary sequence 100000, 100001, 100010, 100011, ... for $F = 5$)

For measuring the similarity between the two probability vectors $x, y \in [0, 1]^F$, the natural loss function would be the cross entropy $\mathrm{H}(x, y)$ or, equivalently, the negative log likelihood.

$$\mathrm{H}(x, y) = \mathbb{E}_x \langle -\log P(x = y) \rangle = \sum_{i=1}^{F} -x_i \log y_i - (1 - x_i) \log(1 - y_i). \tag{1}$$

However, applying it directly to the matrices $X, Y$ unnecessarily limits the global optima of this loss because it does not consider the permutations between $N$ objects, e.g., for $X = [o_1, o_2, o_3]$, $Y = [o_2, o_3, o_1]$ is not the global minima of $\mathrm{H}(X, Y)$. Previous approach (Probst, 2018) tried to solve this problem by learning an additional permutation matrix that "fixes" the order, basically requiring to memorize the ordering.

We take a different approach of directly fixing this loss function. The target objective is to maximize the probability of two sets being equal, thus ideally, at the global minima, two sets $X$ and $Y$ should be equal. Equivalence of two sets is defined as:

$$\begin{aligned} X = Y \Longleftrightarrow & X \subseteq Y \ \wedge \ X \supseteq Y \\ \Longleftrightarrow & (\forall x \in X; x \in Y) \ \wedge \ (\forall y \in Y; y \in X). \end{aligned} \tag{2}$$

However, under the assumption that $|X| = |Y| = N$ and $X$ contains $N$ distinct elements (no duplicates), $X \subseteq Y$ is a sufficient condition for $X = Y$. (Proof: If $X \subseteq Y$ and $X \not\supseteq Y$, there are some $y' \in Y$ such that $y' \notin X$. Since $N$ distinct elements in $X$ are also included in $Y$, $y'$ becomes $Y$'s $N + 1$-th element, which contradicts $|Y| = N$. Note that this proof did not depend on the distinctness of $Y$'s elements.)

Under this condition, therefore,

$$\begin{aligned} X = Y \Longleftrightarrow & X \subseteq Y \\ \Longleftrightarrow & \forall x \in X; x \in Y \\ \Longleftrightarrow & \forall x \in X; \exists y \in Y; x = y \\ \Longleftrightarrow & \bigwedge_{x \in X} \bigvee_{y \in Y} x = y. \end{aligned} \tag{3}$$

We now translate this logical formula into the corresponding log likelihood as follows:

$$\log P(X = Y) = \log P(\bigwedge_{x \in X} \bigvee_{y \in Y} x = y) = \sum_{x \in X} \log P(\bigvee_{y \in Y} x = y) \tag{4}$$

$$= \sum_{x \in X} \log \sum_{y \in Y} P(x = y) \quad \because \text{each } x = y_i \text{ are mutually exclusive.} \tag{5}$$

$$= \sum_{x \in X} \log \sum_{y \in Y} \exp \log P(x = y) \tag{6}$$

$$= \sum_{x \in X} \text{logsumexp}_{y \in Y} \log P(x = y). \tag{7}$$

$$\text{Set Cross Entropy:} \quad \text{SH}(X, Y) \overset{\text{def}}{=} \mathbb{E}_X \langle -\log P(X = Y) \rangle$$

$$= - \sum_{x \in X} \text{logsumexp}_{y \in Y}(-\text{H}(x, y)). \tag{8}$$

$$\because \text{each } x \text{ is independent.}$$

This Set Cross Entropy has the following characteristics: First, compared to the original cross entropy loss, whose global minima is limited to the data point that preserves the same ordering of the elements, SCE increases the number of global minima exponentially by making every permutations of the point also the global minima.

Next, notice that logsumexp is a smooth upper approximation of the maximum, therefore $\text{SH}(X, Y)$ is upper-bounded by the set average equivalent,

$$\text{SH}(X, Y) \leq - \sum_{x \in X} \max_{y \in Y}(-\text{H}(x, y)) = \sum_{x \in X} \min_{y \in Y} \text{H}(x, y) = N \cdot A_{1\text{H}}(X, Y). \tag{9}$$

Intuitively, this is because Eq.9 returns a value which does not account for the possibility that the current closest $y = \arg\min_y \text{H}(x, y)$ of $x$ may not converge to the $x$ in the future during the training.

We illustrate this by comparing two examples: Let $X = \{[0, 1], [0, 0]\}$, $Y_1 = \{[0.1, 0.5], [0.1, 0.5]\}$ and $Y_2 = \{[0.1, 0.5], [0.9, 0.5]\}$. The set cross entropy Eq.8 reports the smaller loss for $\text{SH}(X, Y_1) = -\log 0.81 \approx 0.09$ than for $\text{SH}(X, Y_2) = -\log 0.25 \approx 0.60$. This is reasonable because the global minima is given when the first axis of both $y$s are 0 — $Y_2$ should be more penalized than $Y_1$ for the 0.9 in the second element. In contrast, Eq.9 considers only the closest element ($\arg\min_{y \in Y} \text{H}(x, y) = [0.1, 0.5]$) for each $x$, therefore returns the same loss $= -\log 0.2025 \approx 0.69$ for both cases, ignoring $[0.9, 0.5]$ completely. In fact, Eq.9 has zero gradient at $Y = \{[0, 0.5], [y, 0.5]\}$ for any $y \in [0, 1]$.

Furthermore, the following inequality suggests that the traditional cross entropy between the matrices $X$ and $Y$ is an even looser upper bound of Eq.9. Here, $x_i, y_i$ are the $i$-th element of the vector representation of $X$ and $Y$, respectively:

$$\text{SH}(X, Y) \leq \sum_{x \in X} \min_{y \in Y} \text{H}(x, y) \qquad \because \text{Eq.9}$$

$$\leq \sum_{x_i \in X} \text{H}(x_i, y_i) \qquad \because \forall y_i; \ \min_{y \in Y} \text{H}(x, y) \leq \text{H}(x, y_i)$$

This gives a natural interpretation that ignoring the permutation reduces the cross entropy.

## 4 EVALUATION

### 4.1 OBJECT SET RECONSTRUCTION

The purpose of the task is to obtain the latent representation of a set of objects and reconstruct them, where each object is represented as a feature vector. We prepared two datasets originating from classical AI domains: Sliding tile puzzle (8-puzzle) and Blocksworld.

Learning to reason about the object-based, set representation of the environment is crucial in the robotic systems that continuously receive the list of visible objects from the visual perception module (e.g. Redmon et al. (2016, YOLO)). In a real-world systems, appropriate handling of the set is necessary because it is unnatural to assume that the objects in the environments are always reported in the same order. In particular, the objects even in the *same* environment state may be reported in various orders if multiple such modules are running in parallel in an asynchronous manner.

In this experiment, we show that the permutation invariant loss function like SCE is necessary for learning to reconstruct a set in such a scenario. In this setting, a network is required to reconstruct a set from a single latent representation, while the objects as the target output may be randomly reordered each time the same set is observed and presented to the neural network.

## 8 PUZZLE

Each feature vector as an object consists of 15 features, 9 of which represent the tile number (object ID) and the remaining 6 represent the coordinates. Each data point has 9 such vectors, corresponding to the 9 objects in a single tile configuration. The entire state space of the puzzle is 362880 states. We generated 5000 states and used the 4500 states as the training set.

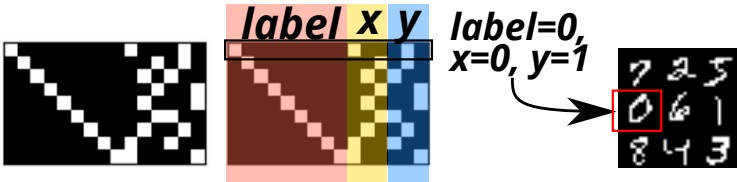

Figure 1: A single 8-puzzle state as a 9x15 matrix, representing 9 objects of 15 features. The first 9 features are the tile numbers and the other 6 features are the 1-hot x/y-coordinates.

We prepared an autoencoder with the permutation invariant layers (Zaheer et al., 2017) as the encoder and the fully-connected layers as the decoder. Since it uses a permutation-invariant encoder, the latent space is already guaranteed to learn a representation that is invariant to the input ordering. The key question here is then whether they can be robustly trained against the random permutations in the training examples for the output.

We tested the reconstruction ability in four scenarios: **(1)** In the first scenario, the dataset is provided in a standard manner. **(2)** In the second scenario, we augment the input dataset by repeating the elements 5 times and randomly reorder the object vectors in each set. The randomized dataset is used as the input to the network, while the target output is still the original dataset (repeated 5 times, without reordering). The purpose of this experiment is to verify the claim of the Deep Set (Zaheer et al., 2017) that it is able to handle the input in a permutation invariant manner. In order to compensate the datasize difference, the maximum training epoch is reduced by 1/5 times compared to the first scenario. **(3)** In the third scenario, we apply the similar operation to the target output of the network. Essentially we always feed the input in the same fixed order while forcing it to learn from the randomized target output. Each time the same data is presented, the target output has the different ordering while the input has the fixed ordering. Therefore, the training should be performed in such a way that the ordering in the output is properly ignored. **(4)** Finally, in the fourth scenario, the ordering in both the input and the output are randomized.

We trained the same network with four different loss functions, **(a)** the traditional cross entropy H, **(b)** Set Cross Entropy SH, **(c)** directed set average of the cross entropy $A_{1H}$ and **(d)** the directed Hausdorff measure of the cross entropy $\mathcal{H}_{1H}$, resulting in 16 training scenarios in total. We performed the same experiment 10 times and took the statistics. The purpose of this is to address the potential concern about the stability of the training. We kept the same set of training/testing data, and the only difference between the runs is the random seed.

We first measured the Set Cross Entropy value between the test dataset and its reconstruction in the above 16 scenarios. Table 1 shows the results. The training with the standard cross entropy loss (H) succeeds in cases **(1,2)** while failed in cases **(3,4)**. The case **(2)** reproduces the claim in (Zaheer et al., 2017) that it encodes the input in an permutation-invariant manner, while it failed in the latter cases because the training is not permutation-invariant with regard to the output. In contrast, the training

with the Set Cross Entropy loss succeeds in all cases. This shows that the permutation-invariant loss function is necessary for training a network with a dataset consisting of sets.

The training with set average distance $A_{1H}$ also reduces the Set Cross Entropy because it is an upper-bound approximation of the Set Cross Entropy. However, in one of the 10 runs, $A_{1H}$ did not converge, showing that the A1H (baseline) could be unstable, possibly due to the issue explained in the example at the end of section 3. In contrast, the training with Hausdorff distance failed to learn the representation at all.

| | Test error in 10 runs (measured by SH) | | | | | | | |
| | Best | | | | Worst | | | |
| Target ordering | Fixed | | Random | | Fixed | | Random | |
| Input ordering | Fixed | Random | Fixed | Random | Fixed | Random | Fixed | Random |
| H | **0.00** | **0.00** | 29.28 | 30.79 | 5.04 | 0.01 | 42.24 | 41.91 |
| SH | **0.00** | **0.00** | **0.00** | **0.00** | 0.15 | **0.03** | 0.10 | 0.07 |
| $A_{1H}$ | **0.00** | **0.00** | **0.00** | **0.00** | **0.14** | 133.34 | **0.09** | **0.00** |
| $\mathcal{H}_{1H}$ | 28.27 | 28.28 | 28.26 | 28.26 | 233.47 | 167.74 | 184.41 | 196.14 |
| | Median | | | | Mean | | | |
| Target ordering | Fixed | | Random | | Fixed | | Random | |
| Input ordering | Fixed | Random | Fixed | Random | Fixed | Random | Fixed | Random |
| H | 0.04 | **0.00** | 32.87 | 32.57 | 0.59 | **0.00** | 34.14 | 33.44 |
| SH | **0.00** | **0.00** | **0.00** | **0.00** | 0.02 | **0.00** | 0.02 | 0.01 |
| $A_{1H}$ | **0.00** | **0.00** | **0.00** | **0.00** | 0.02 | 13.39 | **0.01** | **0.00** |
| $\mathcal{H}_{1H}$ | 31.85 | 28.39 | 31.33 | 28.56 | 77.85 | 59.27 | 50.37 | 67.50 |

Table 1: The summary of test errors out of 10 runs. Best results in **bold**. SH and $A_{1H}$ both succeeded to achieve a good log likelihood sufficiently often. The set average $A_{1H}$ however suffered from a divergence in one training instance, showing its potential instability. The traditional cross entropy H fails to converge when the output is presented in a different order in each iteration. Hausdorff distance failed to converge in all cases.

We next measured the rate of the successful reconstruction among the entire dataset. The "successful reconstruction" is defined as follows: Recall that every data point is a discrete binary vector in the 8-Puzzle dataset while the output of the network is a continuous $N \times F$ matrix of reals between 0 and 1. Therefore, we round the output of the network to $0/1$ and directly compare the result with the input. If every object vector in a set is matched by some of the output object vector, then it is counted as a success. Similar results were obtained in Table 6: SH and $A_{1H}$ both succeeded to achieve a high success rate, while other two metrics completely failed. (We rerun the experiment, therefore the divergence of $A_{1H}$ in the previous experiment did not happen this time.)

Finally, to address the claim that the network is able to learn from the dataset with the variable set size, we performed an experiment which applies the dummy-vector scheme (Sec. 3). In this experiment, we modified the dataset to model such a scenario by randomly dropping one to five elements out of 9 elements. The maximum number of elements is 9. The dropping scheme is specified as follows: Out of the 5000 states generated in total (including the training / testing dataset), approximately half of the states have 9 tiles, $1/4$ of the states have 8 tiles, ... and $1/2^5$ of the states have 5 tiles. The elements to drop are selected randomly.

The results in Table 3 shows that the training with our proposed SH loss function achieves the best success ratio for the reconstruction. The reconstruction includes the dummy vectors, indicating that the network is able to represent not only the elements in the set but also the number of the missing elements.

### BLOCKSWORLD

In order to test the reconstruction ability for the more complex feature vectors, we prepared a photo-realistic Blocksworld dataset (Fig. 2) which contains the blocks world states rendered by Blender 3D engine. There are several cylinders or cubes of various colors and sizes and two surface materials (Metal/Rubber) stacked on the floor, just like in the usual STRIPS (McDermott, 2000) Blocksworld domain. In this domain, three actions are performed: `move` a block onto another stack or on the

| | Reconstruction success ratio in 10 runs | | | | | | | |
| --- | --- | --- | --- | --- | --- | --- | --- | --- |
| | Best | | | | Worst | | | |
| Target ordering | Fixed | | Random | | Fixed | | Random | |
| Input ordering | Fixed | Random | Fixed | Random | Fixed | Random | Fixed | Random |
| H | 0.00 | 0.00 | 0.00 | 0.00 | 0.00 | 0.00 | 0.00 | 0.00 |
| SH | **1.00** | **1.00** | **1.00** | **1.00** | **1.00** | **1.00** | 0.89 | **1.00** |
| $A_{1H}$ | **1.00** | **1.00** | **1.00** | **1.00** | **1.00** | **1.00** | **1.00** | **1.00** |
| $\mathcal{H}_{1H}$ | 0.00 | 0.00 | 0.00 | 0.00 | 0.00 | 0.00 | 0.00 | 0.00 |
| | Median | | | | Mean | | | |
| Target ordering | Fixed | | Random | | Fixed | | Random | |
| Input ordering | Fixed | Random | Fixed | Random | Fixed | Random | Fixed | Random |
| H | 0.00 | 0.00 | 0.00 | 0.00 | 0.00 | 0.00 | 0.00 | 0.00 |
| SH | **1.00** | **1.00** | **1.00** | **1.00** | **1.00** | **1.00** | 0.99 | **1.00** |
| $A_{1H}$ | **1.00** | **1.00** | **1.00** | **1.00** | **1.00** | **1.00** | **1.00** | **1.00** |
| $\mathcal{H}_{1H}$ | 0.00 | 0.00 | 0.00 | 0.00 | 0.00 | 0.00 | 0.00 | 0.00 |

Table 2: The summary of the success rate for the 10 runs of 16 training scenarios. Best results in **bold**. SH and $A_{1H}$ both succeeded to reconstruct the binary vectors in the 8 puzzles. The traditional cross entropy H and the Hausdorff distance $\mathcal{H}_{1H}$ both failed to reconstruct the binary vectors.

| | Reconstruction success ratio in 10 runs | | | | | | | |
| --- | --- | --- | --- | --- | --- | --- | --- | --- |
| | Best | | | | Worst | | | |
| Target ordering | Fixed | | Random | | Fixed | | Random | |
| Input ordering | Fixed | Random | Fixed | Random | Fixed | Random | Fixed | Random |
| H | 0.03 | 0.49 | 0.05 | 0.56 | 0.00 | 0.00 | 0.00 | 0.00 |
| SH | **0.62** | **0.63** | **0.65** | **0.65** | **0.52** | **0.54** | **0.57** | **0.54** |
| $A_{1H}$ | **0.62** | 0.62 | 0.60 | 0.59 | **0.52** | 0.09 | 0.51 | 0.50 |
| $\mathcal{H}_{1H}$ | 0.00 | 0.00 | 0.00 | 0.00 | 0.00 | 0.00 | 0.00 | 0.00 |
| | Median | | | | Mean | | | |
| Target ordering | Fixed | | Random | | Fixed | | Random | |
| Input ordering | Fixed | Random | Fixed | Random | Fixed | Random | Fixed | Random |
| H | 0.00 | 0.12 | 0.00 | 0.27 | 0.01 | 0.17 | 0.01 | 0.31 |
| SH | 0.57 | **0.59** | **0.60** | **0.60** | 0.58 | **0.59** | **0.61** | **0.60** |
| $A_{1H}$ | **0.59** | 0.57 | 0.57 | 0.56 | 0.58 | 0.53 | 0.57 | 0.56 |
| $\mathcal{H}_{1H}$ | 0.00 | 0.00 | 0.00 | 0.00 | 0.00 | 0.00 | 0.00 | 0.00 |

Table 3: The summary of the success rate for the 10 runs of 16 training scenarios, where the size of the set randomly varies from 4 to 9 in the dataset. Best results in **bold**. The proposed SH achieved the best success rate overall, $A_{1H}$ comes next, the traditional cross entropy H and the Hausdorff distance $\mathcal{H}_{1H}$ both failed in most cases.

floor, and `polish`/`unpolish` a block i.e. change the surface of a block from Metal to Rubber or vice versa. All actions are applicable only when the block is on top of a stack or on the floor. The latter actions allow changes in the non-coordinate features of the object vectors.

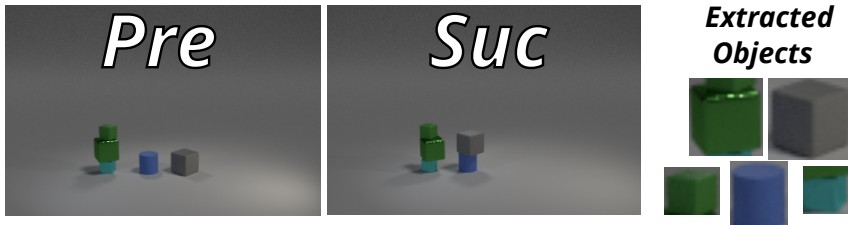

Figure 2: An example Blocksworld transition. Each state has a perturbation from the jitter in the light positions and the ray-tracing noise. Objects have the different sizes, colors, shapes and surface materials. Regions corresponding to each object in the environment are extracted according to the bounding box information included in the dataset generator output, but is ideally automatically extracted by object recognition methods such as YOLO (Redmon et al., 2016). Other objects may intrude the extracted regions.

The dataset generator produces a 300x200 RGB image and a state description which contains the bounding boxes (bbox) of the objects. Extracting these bboxes is a object recognition task we do not address in this paper, and ideally, should be performed by a system like YOLO (Redmon et al., 2016). We resized the extracted image patches in the bboxes to 32x32 RGB, compressed it into a feature vector of 1024 dimensions with a convolutional autoencoder, then concatenated it with the bbox $(x_1, y_1, x_2, y_2)$ which is discretized by 5 pixels and encoded as 1-hot vectors (60/40 categories for $x/y$-axes), resulting in 1224 features per object. The generator is able to enumerate all possible states (80640 states for 5 blocks and 3 stacks). We used 2250 states as the training set and 250 states as the test set.

We also verified the results qualitatively. Some reconstruction results are visualized in Fig. 3. These visualizations are generated by pasting the image patches decoded from the first 1024 axes of the reconstructed 1224-D feature vectors in a position specified by the reconstructed bounding box in the last 200 axes.

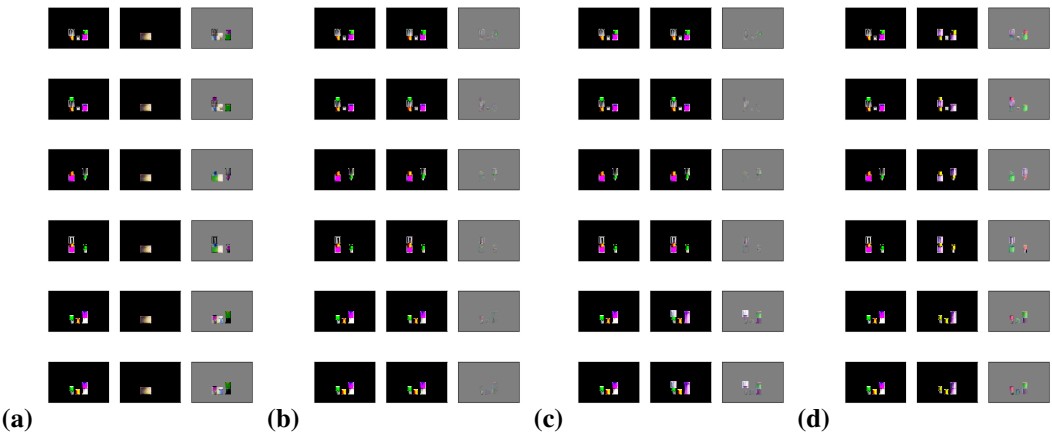

Figure 3: The visualizations of the Blocksworld state input (left), its reconstruction (middle) and their pixel-wise difference (right). From the left, each three columns represent **(a)** the traditional cross entropy H, **(b)** Set Cross Entropy SH, **(c)** directed set average of the cross entropy $A_{1H}$ and **(d)** the directed Hausdorff measure of the cross entropy $\mathcal{H}_{1H}$. The proposed **(b)** Set Cross Entropy correctly reconstructs the input.

Results in Table 4 shows that the training with SH and $A_{1H}$ achieved a better test error compared to the other metrics.

| | Best test error in 10 runs (measured by SH) | | | |
|---|---|---|---|---|
| Target ordering | Fixed | | Random | |
| Input ordering | Fixed | Random | Fixed | Random |
| H | 3360.22 | 3360.26 | 3425.89 | 3434.60 |
| SH | **3253.70** | 3260.04 | **3251.32** | **3252.71** |
| $A_{1H}$ | 3258.84 | **3251.74** | 3261.13 | 3264.82 |
| $\mathcal{H}_{1H}$ | 3409.58 | 3410.77 | 3415.18 | 3373.22 |

Table 4: The best results of 10 runs. Both SH and $A_{1H}$ successfully converged below the sufficient accuracy.

In Table 5, as another metric with the more intuitive sense, we measure the difference between the visualization results of the input and the output (as shown in Fig. 3) by the Root Mean Squared Error of the pixel values averaged over RGB, pixels and the dataset. Each pixel is represented in the $[0, 1]$ range (closed set of reals between 0 and 1), thus the 0.1 on the table means that pixels differ by 0.1 on average.

| | RMSE between the visualized image | | | |
|---|---|---|---|---|
| Target ordering | Fixed | | Random | |
| Input ordering | Fixed | Random | Fixed | Random |
| H | 0.10 | 0.10 | 0.15 | 0.15 |
| SH | **0.08** | **0.08** | **0.07** | **0.08** |
| $A_{1H}$ | **0.08** | 0.10 | **0.08** | **0.08** |
| $\mathcal{H}_{1H}$ | 0.14 | 0.15 | 0.16 | 0.15 |

Table 5: The best results of 10 runs. Both SH and $A_{1H}$ successfully converged below the sufficient accuracy.

## 4.2 RULE LEARNING ILP TASKS

The purpose of this task is to learn to generate the prerequisites (body) of the first-order-logic horn clauses from the head of the clause. Unlike the previous tasks, this task is not an autoencoding task. The bodies are considered as a set because the order of the terms inside a body does not matter for the clause to be satisfied.

The main purpose of this experiment is to show the effectiveness of our approach on set generation, not to demonstrate a more general neural theorem proving system. An interesting avenue of future work is to see how our approach can help the existing work on neural theorem proving.

We used a Countries dataset (Bouchard et al., 2015) that contains 163 countries and trained the models for $n$-hop neighbor relations. For example, for $n = 2$, given a head neighbor2(austria, germany, belgium) as an input, the task is to predict the body {neighborOf(austria, germany), neighborOf(germany, belgium)}, which is a set of two terms. This is a weaker form of a more general backward chaining used in Neural Theorem Proving (Rocktäschel & Riedel, 2017) because the output does not contain free variables.

In the $n$-neighbor scenario, the input is a $2 + 163(n + 1)$-dimensional vector, which consists of a one-hot label of 2 categories for the predicate of the head, and $n + 1$ one-hot labels of 163 categories for the arguments of the head. For example, a head neighbor2(austria, germany, belgium) spends 2 dimensions for identifying the predicate neighbor2, and three 1-hot vectors of 163 categories for representing austria,germany,belgium. The output is a $n \times 328$ matrix, where each row represents a binary predicate ($328 = 2 + 2 \cdot 163$). This is again because the answer is {neighborOf(austria, germany), neighborOf(germany, belgium)}: There are 2 elements in the set, thus the output is a $2 \times 328$ matrix. Each element uses 2 dimensions for identifying the predicate head neighborOf and two 1-hot vectors of 163 categories for the arguments (e.g. austria and germany).

We trained the network with the neighbor-$n$ datasets ranging from $n = 2$ to $n = 5$ (see the result table for the detailed domain characteristics). The softmax output of the network is parsed back to the

symbolic representation by selecting the index that gives the maximum probability, then compared against the test examples as a set. We counted the ratio of the clauses across the test set where every body term matches against one of the output terms. The output data (body terms) may have an arbitrary ordering, and we have another variant similar to the previous experiment: In the randomized body order dataset, the dataset is repeated 5 times, while the ordering of the terms inside each body is randomly shuffled.

Table 6 shows that the network with Set Cross Entropy achieved the best accuracy, set average generally comes in the second and the traditional cross entropy struggles. This trend was observed not only in the the randomized-body-ordering dataset, which observes the same body in a different order in each iteration, but also in the fixed-body-ordering dataset. This shows that the Set Cross Entropy relaxes the search space by adding more global minima and making the training easier.

## 5 DISCUSSION

Vinyals et al. (2016) repeatedly emphasized the advantage of limiting the possible equivalence classes of the outputs by engineering the training data for solving the combinatorial problems. For example, they pre-sorted the training example for the Delaunay triangulation (set of triangles) by the lexicographical order and trained an LSTM model with the standard cross entropy (Vinyals et al., 2015). However, this is an ad-hoc method that depends on the particular domain knowledge and, as we have shown, the difficulty of learning such an output was caused by the loss function that considers the ordering. Moreover, we showed that the standard cross entropy and the set average metrics are the less tighter upper bound of the proposed Set Cross Entropy and also that it empirically outperforms the standard cross entropy in the theory learning task, even if a specific ordering is imposed on the output.

One limitation of the current approach is that the set cross entropy contains a double-loop, therefore takes $O(N^2)$ runtime for a set of $N$ objects. However, unlike the algorithm proposed in Probst (2018), which uses a sequential Gale-Shapley algorithm which also uses $O(N^2)$ runtime, our loss function can be efficiently implemented on GPUs because it consists of a simple combination of logsumexp and summation.

Still, improving the runtime complexity is an important direction for future work because the other set reconstruction tasks, including 3D point clouds datasets like Shapenet (Chang et al., 2015), may contain a much larger number of elements in each set. A promising candidate for tacking this difficulty is to combine Set Cross Entropy with Approximate $k$-Nearest Neighbor (Indyk & Motwani, 1998) methods, especially the Locality Sensitive Hashing (Wang et al., 2016, LSH). LSH can preprocess and divide the target output $X$ into the subsets within a certain radius and we can limit the inner loop to each subset. The resulting method can be seen as the midpoint of Set Cross Entropy and set average because set average (Eq.9) is the special case of this extension that uses the nearest neighbor ($\min_{y \in Y} H(x, y)$) and worked reasonably well in the tasks evaluated in this paper. The main obstacle for this approach would be to extend Set Cross Entropy to a metric variant that satisfies the metric axioms (non-negativity, identity, symmetry, the triangular inequality) that is required for LSH methods in general. One candidate in this direction is a variant of Jensen-Shannon divergence called S2JSD (Endres & Schindelin, 2003), which satisfies the metric axioms and has a LSH method (Mao et al., 2017).

Another direction for future work is to use the Long Short Term Memory (Hochreiter & Schmidhuber, 1997) for handling the sets without imposing the shared upper bound on the number of elements in a set, which has been already explored in the literature (Vinyals et al., 2015; 2016). Since our approach is agnostic to the type of the neural network, they are orthogonal to our approach.

## 6 CONCLUSION

In this paper, we proposed Set Cross Entropy, a measure that models the likelihood between the sets of probability distributions. When the output of the neural network model can be naturally regarded as a set, Set Cross Entropy is able to relax the search space by making the permutations of a global minima also the global minima, and makes the training easier. This is in contrast to the existing approaches that try to correct the ordering of the output by learning a permutation matrix, or an

| | The rate of correct answering on the test set, 10 runs | | | | | | | |
|---|---|---|---|---|---|---|---|---|
| | $n = 2$, neighbor2$(a, b, c)$:-neighborOf$(a, b)$, neighborOf$(b, c)$ | | | | | | | |
| | Dataset: 2858 ground clauses; Training: 2250 clauses; Test: 250 clauses. | | | | | | | |
| Target ordering | Fixed | | | | Random | | | |
| | Best | Worst | Median | Mean | Best | Worst | Median | Mean |
| H | 0.32 | 0.23 | 0.29 | 0.28 | 0.36 | 0.25 | 0.31 | 0.31 |
| SH | **0.96** | **0.90** | **0.91** | **0.92** | **0.94** | **0.88** | **0.91** | **0.91** |
| $A_{1H}$ | 0.91 | 0.79 | 0.86 | 0.86 | **0.94** | 0.72 | 0.88 | 0.87 |
| $\mathcal{H}_{1H}$ | 0.87 | 0.66 | 0.82 | 0.80 | 0.85 | 0.66 | 0.83 | 0.81 |
| | $n = 3$, neighbor3$(a, b, c, d)$:- ... | | | | | | | |
| | Dataset: 11000 ground clauses; Training: 2250 clauses; Test: 250 clauses. | | | | | | | |
| Target ordering | Fixed | | | | Random | | | |
| | Best | Worst | Median | Mean | Best | Worst | Median | Mean |
| H | 0.10 | 0.04 | 0.06 | 0.06 | 0.10 | 0.06 | 0.07 | 0.07 |
| SH | **0.72** | **0.55** | **0.61** | **0.62** | **0.70** | 0.53 | **0.66** | **0.64** |
| $A_{1H}$ | 0.61 | 0.52 | 0.55 | 0.56 | 0.60 | **0.53** | 0.57 | 0.57 |
| $\mathcal{H}_{1H}$ | 0.55 | 0.31 | 0.44 | 0.44 | 0.57 | 0.37 | 0.43 | 0.45 |
| | $n = 4$, neighbor4$(a, b, c, d, e)$:- ... | | | | | | | |
| | Dataset: 39878 ground clauses; Training: 2250 clauses; Test: 250 clauses. | | | | | | | |
| Target ordering | Fixed | | | | Random | | | |
| | Best | Worst | Median | Mean | Best | Worst | Median | Mean |
| H | 0.02 | 0.00 | 0.01 | 0.01 | 0.03 | 0.00 | 0.02 | 0.02 |
| SH | **0.38** | **0.24** | **0.33** | **0.32** | **0.36** | **0.28** | **0.32** | **0.32** |
| $A_{1H}$ | 0.33 | 0.22 | 0.27 | 0.26 | 0.34 | 0.22 | 0.26 | 0.26 |
| $\mathcal{H}_{1H}$ | 0.22 | 0.12 | 0.18 | 0.18 | 0.24 | 0.13 | 0.18 | 0.18 |
| | $n = 4$, neighbor4$(a, b, c, d, e)$:- ... | | | | | | | |
| | Dataset: 39878 ground clauses; Training: 9000 clauses; Test: 1000 clauses. | | | | | | | |
| Target ordering | Fixed | | | | Random | | | |
| | Best | Worst | Median | Mean | Best | Worst | Median | Mean |
| H | 0.04 | 0.03 | 0.04 | 0.03 | 0.04 | 0.02 | 0.03 | 0.03 |
| SH | **0.87** | **0.81** | **0.82** | **0.83** | **0.86** | **0.77** | **0.82** | **0.82** |
| $A_{1H}$ | 0.81 | 0.71 | 0.77 | 0.76 | 0.79 | 0.59 | 0.73 | 0.72 |
| $\mathcal{H}_{1H}$ | 0.50 | 0.12 | 0.40 | 0.37 | 0.53 | 0.24 | 0.42 | 0.41 |
| | $n = 5$, neighbor5$(a, b, c, d, e, f)$:- ... | | | | | | | |
| | Dataset: 137738 ground clauses; Training: 2250 clauses; Test: 250 clauses. | | | | | | | |
| Target ordering | Fixed | | | | Random | | | |
| | Best | Worst | Median | Mean | Best | Worst | Median | Mean |
| H | 0.00 | 0.00 | 0.00 | 0.00 | 0.01 | 0.00 | 0.00 | 0.00 |
| SH | **0.17** | **0.10** | **0.11** | **0.12** | **0.15** | **0.06** | **0.11** | **0.11** |
| $A_{1H}$ | 0.13 | 0.06 | 0.10 | 0.10 | 0.13 | **0.06** | **0.11** | 0.10 |
| $\mathcal{H}_{1H}$ | 0.06 | 0.01 | 0.04 | 0.04 | 0.06 | 0.04 | 0.05 | 0.05 |
| | $n = 5$, neighbor5$(a, b, c, d, e, f)$:- ... | | | | | | | |
| | Dataset: 137738 ground clauses; Training: 9000 clauses; Test: 1000 clauses. | | | | | | | |
| Target ordering | Fixed | | | | Random | | | |
| | Best | Worst | Median | Mean | Best | Worst | Median | Mean |
| H | 0.01 | 0.00 | 0.00 | 0.00 | 0.01 | 0.00 | 0.01 | 0.01 |
| SH | **0.65** | **0.52** | **0.55** | **0.56** | **0.60** | **0.50** | **0.56** | **0.55** |
| $A_{1H}$ | 0.53 | 0.46 | 0.48 | 0.48 | 0.56 | 0.46 | 0.50 | 0.50 |
| $\mathcal{H}_{1H}$ | 0.20 | 0.04 | 0.11 | 0.11 | 0.18 | 0.07 | 0.13 | 0.13 |

Table 6: The summary of the rule learning task, 10 runs.

ad-hoc methods that reorder the dataset using the domain-specific expert knowledge. Training based on the Set Cross Entropy is also robust against the dataset which contains vectors whose internal ordering may change time to time in an arbitrary manner. We demonstrated the effectiveness of the approach by comparing Set Cross Entropy against the normal cross entropy, as well as the other set-based metrics such as Hausdorff distance or set average (Chamfer) distance. set average distance was shown to upper-bound Set Cross Entropy, and while it performed comparably well in the object reconstruction task, it was outperformed by Set Cross Entropy in the rule learning task. Training a neural network with Hausdorff distance turned out to be particularly hard, and it failed in many scenarios, showing that it is not suitable as a loss function for the set reconstruction tasks considered in this paper.

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

# APPENDIX

## 6.1 NETWORK MODEL FOR 8 PUZZLE

As mentioned in the earlier sections, the network has a permutation invariant encoder and the fully-connected decoder.

The input to the network is a $9 \times 15$ matrix, where the first dimension represents the objects and the second dimension represents the features of each object. The encoder has two 1D convolution layers of 1000 neurons with filter size 1, modeling the element-wise network $\rho$. The output of these layers is then aggregated by taking the sum of the first dimension. The result is then fed to two another fully-connected layers of width 1000, which maps to the latent layer of 100 neurons. All encoder layers are activated by ReLU. The latent representation is regularized and activated by Gumbel-Softmax Maddison et al. (2017); Jang et al. (2017) as the input is a categorical model.

The decoder consists of three fully-connected layers with dropout and batch normalization as shown below:

$$\begin{vmatrix} \text{fc(1000), relu, batchnorm, dropout(0.5),} \\ \text{fc(1000), relu, batchnorm, dropout(0.5),} \\ \text{dense(135), reshape}(9 \times 15) \end{vmatrix}$$

The last layer is then split into $9 \times 9$, $9 \times 3$, $9 \times 3$ matrices and separately activated by softmax, reflecting the input dataset (Fig. 1).

## 6.2 NETWORK MODEL FOR BLOCKSWOLRD

The same network as the 8-puzzle was used, except that the input and the output is a $5 \times 1224$ matrix. The activations of the last layer is different: The first 1024 features are activated by a sigmoid function, while the 200 features are divided into 40, 60, 40, 60 dimensions (for the one-hot bounding box information) and are separately activated by softmax.

## 6.3 FEATURE EXTRACTION FOR BLOCKSWORLD

The 32x32 RGB image patches in the Blocksworld states are compressed into the feature vectors that are later used as the input. The image features are learned by a convolutional autoencoder depicted in Fig. 4 and Fig. 5.

$$\begin{vmatrix} \text{Input}(32 \times 32 \times 3), \\ \text{GaussianNoise(0.1),} \\ \text{conv2d(filter=16, kernel=}3 \times 3,), \\ \text{relu,} \\ \text{MaxPooling2d}(2 \times 2), \\ \text{conv2d(filter=16, kernel=}3 \times 3,), \\ \text{relu,} \\ \text{MaxPooling2d}(2 \times 2), \\ \text{conv2d(filter=16, kernel=}3 \times 3,), \\ \text{sigmoid} \end{vmatrix}$$

Figure 4: The implementation of the encoder for feature selection, which outputs a $8 \times 8 \times 16$ tensor.

$$\begin{vmatrix} \text{Input}(8 \times 8 \times 16), \\ \text{conv2d(filter=16, kernel=}3 \times 3,), \\ \text{relu,} \\ \text{UpSampling2d}(2 \times 2), \\ \text{conv2d(filter=16, kernel=}3 \times 3,), \\ \text{relu,} \\ \text{UpSampling2d}(2 \times 2), \\ \text{conv2d(filter=16, kernel=}3 \times 3,), \\ \text{relu,} \\ \text{fc(3072),} \\ \text{sigmoid,} \\ \text{reshape}(32 \times 32 \times 3) \end{vmatrix}$$

Figure 5: The implementation of the decoder for feature selection.

## 6.4 NETWORK MODEL FOR RULE LEARNING

We removed the encoder and the latent layer from the above models, and connected the input directly to the decoder. While the decoder has the same types of layers, the width is shrinked to 400. In the $n$-neighbor scenario, the input is a $2 + 163(n + 1)$ vector, which consists of a one-hot label of 2 categories for the predicate of the head, and $n + 1$ one-hot labels of 163 categories for the arguments of the head. 163 categories corresponds to the number of countries in the Countries dataset (Bouchard et al., 2015). The output is a $[n, 328]$ matrix, where each row represents a binary predicate ($328 = 2 + 2 \cdot 163$).

## 6.5 EXAMPLE APPLICATION OF THE PERMUTATION-INVARIANT REPRESENTATION & RECONSTRUCTION

To address the practical utility of "set reconstruction" or "set autoencoding", we added a new experiment. We modified Latplan (Asai & Fukunaga, 2018) neural-symbolic classical planning system, a system that operates on a discrete symbolic latent space of the real-valued inputs and runs Dijkstra's/A* search using a state-of-the-art symbolic classical planning solver. We modified Latplan to take the set-of-object-feature-vector input rather than images. It is a high-level task planner (unlike motion planning / actuator control) that has implications on robotic systems, which perceives a set of inputs already preprocessed by the external system. For example, the image input is first fed into Object Recognition system (e.g. YOLO, (Redmon et al., 2016)) and the planner receives a set of feature vectors extracted from the image patches segmented from the raw image, rather than feeding the image input directly to the planning system.

Latplan system learns the binary latent space of an arbitrary raw input (e.g. images) with a Gumbel-Softmax variational autoencoder, learns a discrete state space from the transition examples, and runs a symbolic, systematic search algorithm such as Dijkstra or A* search which guarantee the optimality of the solution. Unlike RL-based planning systems, the search agent does not contain the learning aspects. The discrete plan in the latent space is mapped back to the raw image visualization of the plan execution, which requires the reconstruction capability of (V)AE. A similar system replacing Gumbel Softmax VAE with Causal InfoGAN was later proposed (Kurutach et al., 2018).

We replaced Latplan's Gumbel-Softmax VAE with our autoencoder used in the 8-Puzzle and the Blocksworld experiments (Appendix, Sec. 6.1, Sec. 6.2). Our autoencoder also uses Gumbel Softmax in the latent layer, but it uses (Zaheer et al., 2017) encoder and is trained with Set Cross Entropy.

When the network learned the representation, it guarantees that the planner finds a solution because the search algorithm being used (e.g. Dijkstra) is a complete, systematic, symbolic search algorithm, which guarantees to find a solution whenever it is reachable in the state space. If the network cannot learn the permutation-invariant representation, the system cannot solve the problem and/or return the human-comprehensive visualization. This makes the specific permutation-invariant representation using (Zaheer et al., 2017) and the proposed Set Cross Entropy necessary when the input is given as a set of future vectors.

## 8 PUZZLE

First, the training was performed on a dataset in which the object vector ordering is randomized. The autoencoder compresses the $15 \times 9 = 135$-bit binary representation (object vectors) into a permutation-invariant 100-bit discrete latent binary representation. We provided 5000 states for training the autoencoder, while the search space consists of 362880 ($= 9!$) states and 967680 transitions.

Note that each state have 9! variations due to the permutations in the order the tiles and the locations are reported. This also increases the number of transition quadratically ($(9!)^2$).

We generated 40 problem instances of 8-puzzle each generated by a random walk from the goal state. 40 instances consist of 20 instances each generated by a 7-steps random walk and another 20 by 14 steps. We solved 40 instances using Fast Downward classical planner Helmert (2004) with blind heuristics in order to remove the effect of heuristics.

We compared the number of problems successfully solved by two variations of Latplan where each uses the autoencoder trained with Set Average and Set Cross Entropy, respectively, for encoding the input into binary latent space. Both version managed to solve all instances because both Set Average and Set Cross Entropy managed to train the AE from 5000 examples with a sufficient accuracy. All solutions were correct (checked manually). Since the search algorithm being used is optimal, the quality of the solution was also identical.

## BLOCKSWORLD

We solved 30 planning instances in a 4-blocks, 3-stacks environment. The instances are generated by taking a random initial state and choosing a goal state by the 3, 7, or 14 steps random walks (10 instances each). The correctness of the plans are again checked manually. The same planner configuration was used for all instances.

The search space consists of 5760 states and 34560 transitions, and each state have 4! variations due to permutations of 4 blocks. We provided 1000 randomly selected states for training the autoencoder.

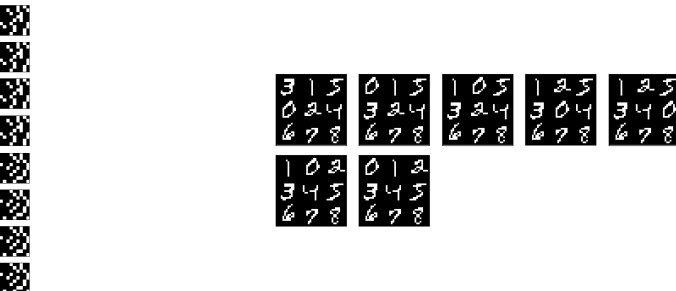

Figure 6: (**Left**) An example plan in a set-of-object-vector form, decoded from its binary latent representation using the permutation-invariant autoencoder (the plan is executed from top to bottom). (**Right**) Its visualization using the tile images (taken from MNIST) pasted onto a black canvas (The plan is executed from left to right, top to bottom).

We compared the number of problems successfully solved by Latplan between two variations of Latplan using the autoencoder trained with Set Average and Set Cross Entropy, respectively. For the total of 30 instances, both Latplan+Set Avg and Latplan+SCE returned plans, however the plans returned by Latplan+Set Avg were correct in 11 instnaces, while Latplan+SCE returned 14 correct instances (Details in Table 7).

As the autoencoder trained by Set Average had larger reconstruction error, it sometimes fails to capture the essential feature of the input, causing the system to return an invalid plan. The common error was changing the surface of the blocks or swapping the blocks without a proper action needed, e.g., moving more than two blocks, move a block and polish another block in a single time step, etc.

| Random walk steps used for generating the problem instances | The number of solved instances (out of 10 instances each) | |
|---|---|---|
| | SH | $A_{1\text{H}}$ |
| 3 | 7 | 7 |
| 7 | **5** | 3 |
| 14 | **2** | 1 |

Table 7: The number of instances solved by Latplan using a VAE trained by Set Cross Entropy (SH) and Set Average ($A_{1\text{H}}$) of the cross entropy.

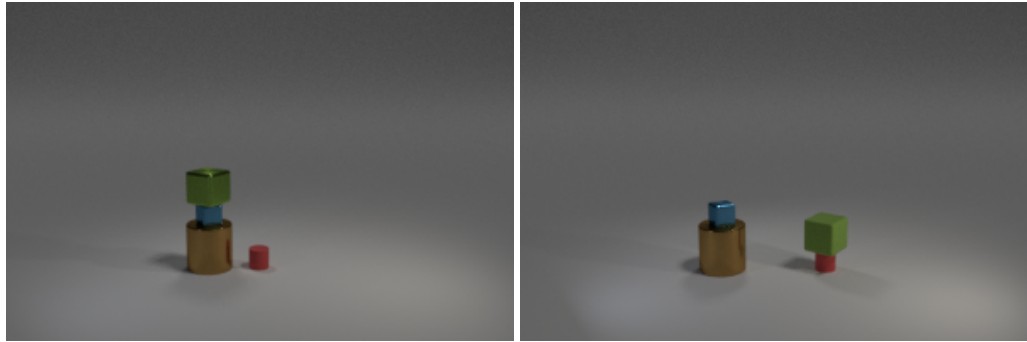

Figure 7: An example of a problem instance. (**Left**) The initial state. (**Right**) The goal state. The planner should unpolish a green cube and move the blocks to the appropriate goal position, while also following the environment constraint that the blocks can move or polished only when it is on top of a stack (including the floor itself).

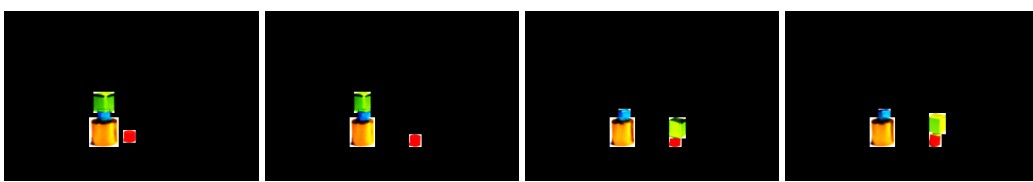

Figure 8: An example of a successful plan execution, returned by Latplan using the AE trained by the proposed Set Cross Entropy method. The AE is used for encoding the object-vector input into a binary space that is suitable for Dijkstra search. While the problem was generated by a 7-step random walk from the goal state, Latplan found a shorter, optimal solution because of the underlying optimal search algorithm (Dijkstra).

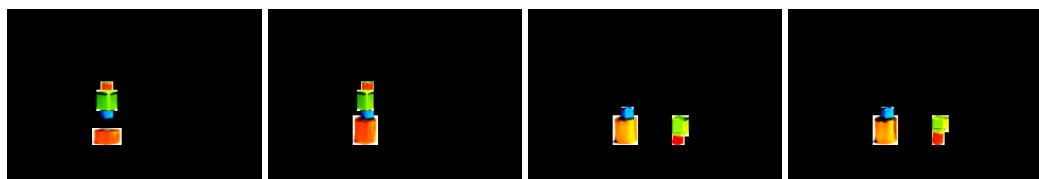

Figure 9: The decoded solution found by Latplan for the same instance, where the AE is trained by SetAverage, which had a higher mean square error for the reconstruction. As a result, not only the initial state is invalid, but also, at the second step, two blocks are simultaneously moved in a single action, which is an invalid state transition.

## 6.6 ENTIRE PLANNING RESULTS

## 8 PUZZLE

Problem 000, generated by 007 steps

Set Cross Entropy

Set Average

Visualization

Visualization

Problem 001, generated by 007 steps

Set Cross Entropy

Set Average

Visualization

Visualization

Problem 002, generated by 007 steps

Set Cross Entropy                    Set Average

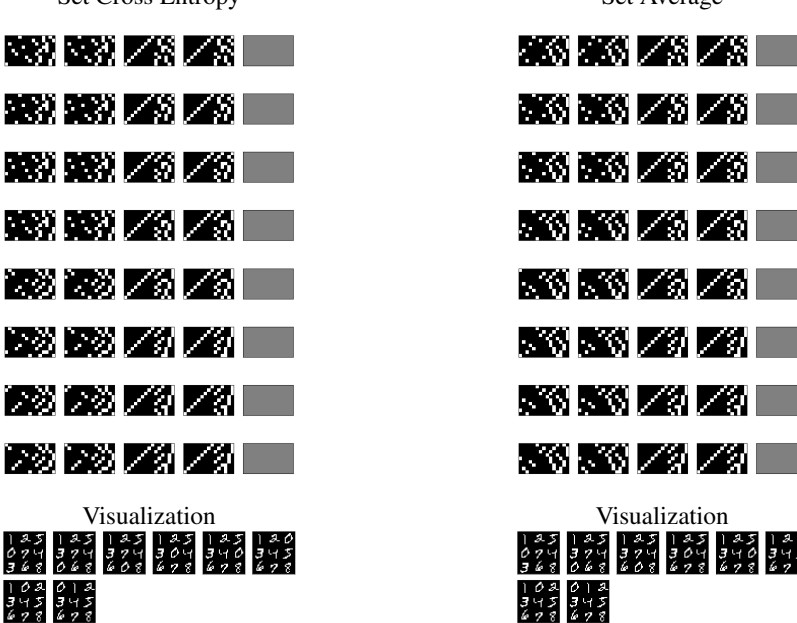

Visualization                    Visualization

Problem 003, generated by 007 steps

Set Cross Entropy                    Set Average

Visualization                    Visualization

Problem 004, generated by 007 steps

Set Cross Entropy

Set Average

Visualization

Visualization

Problem 005, generated by 007 steps

Set Cross Entropy

Set Average

Visualization

Visualization

Problem 006, generated by 007 steps

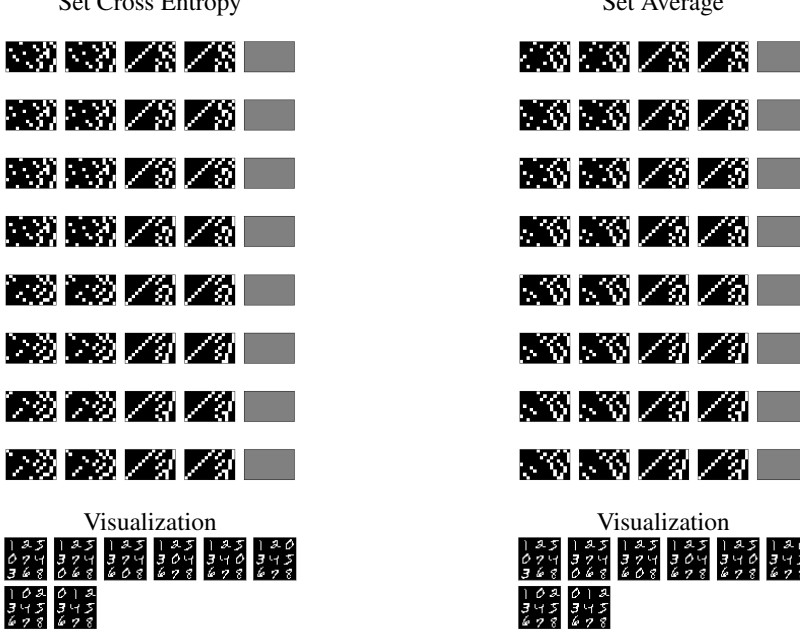

Problem 007, generated by 007 steps

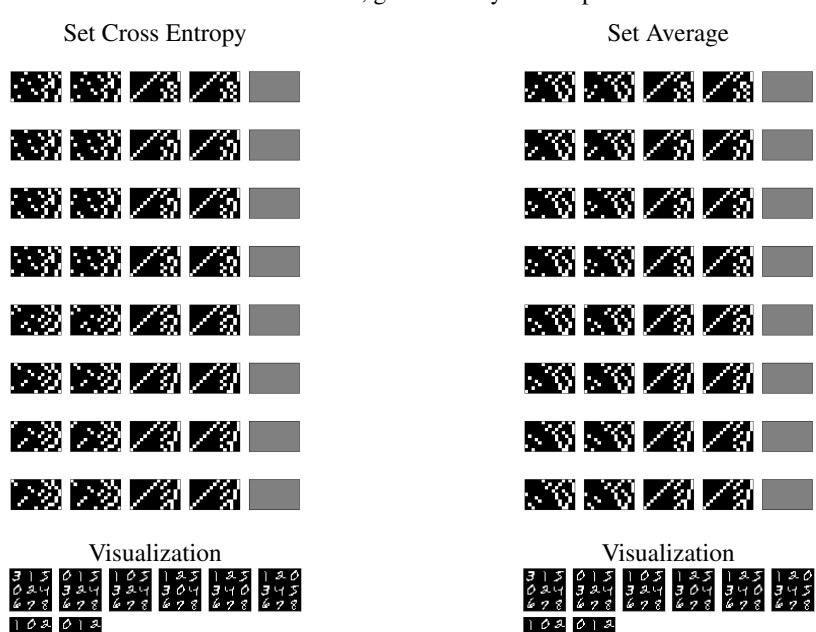

Problem 008, generated by 007 steps

Set Cross Entropy                    Set Average

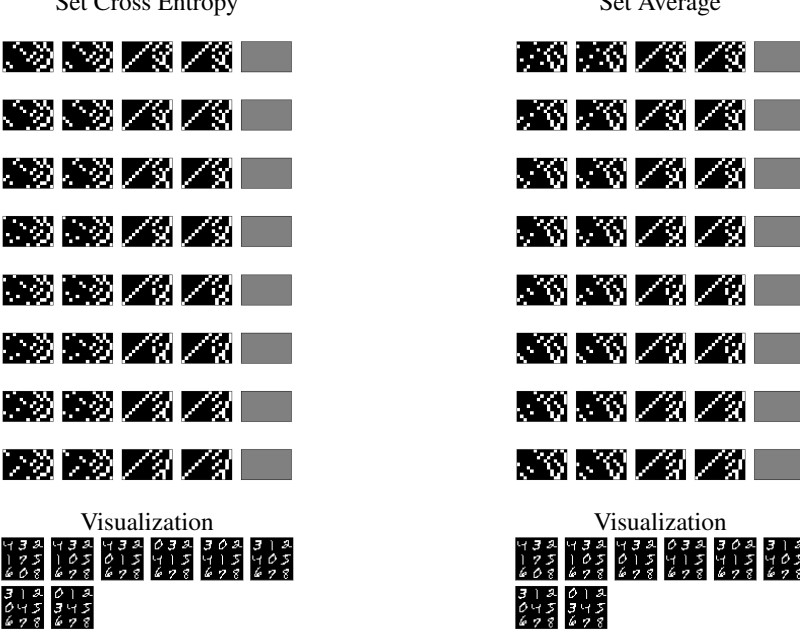

Visualization                    Visualization

Problem 009, generated by 007 steps

Set Cross Entropy                    Set Average

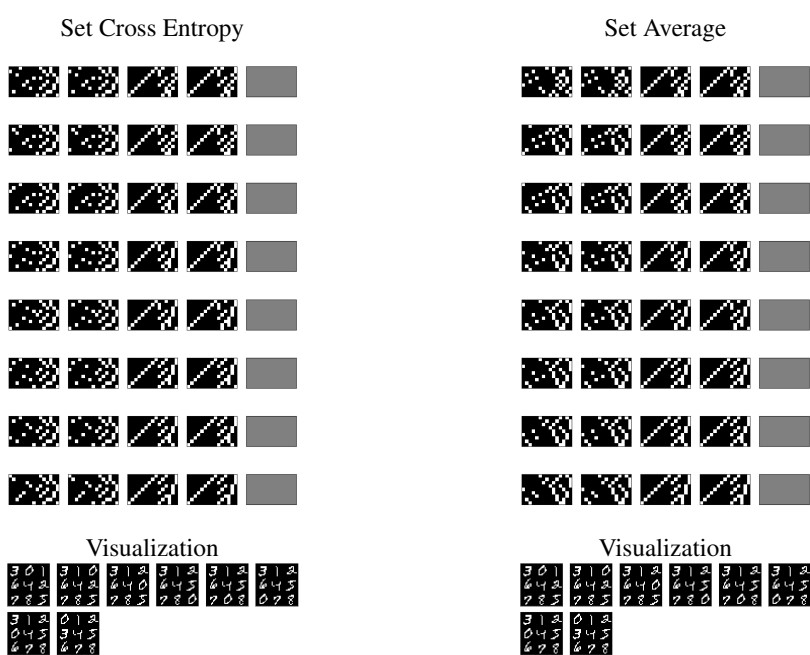

Visualization                    Visualization

Problem 000, generated by 014 steps

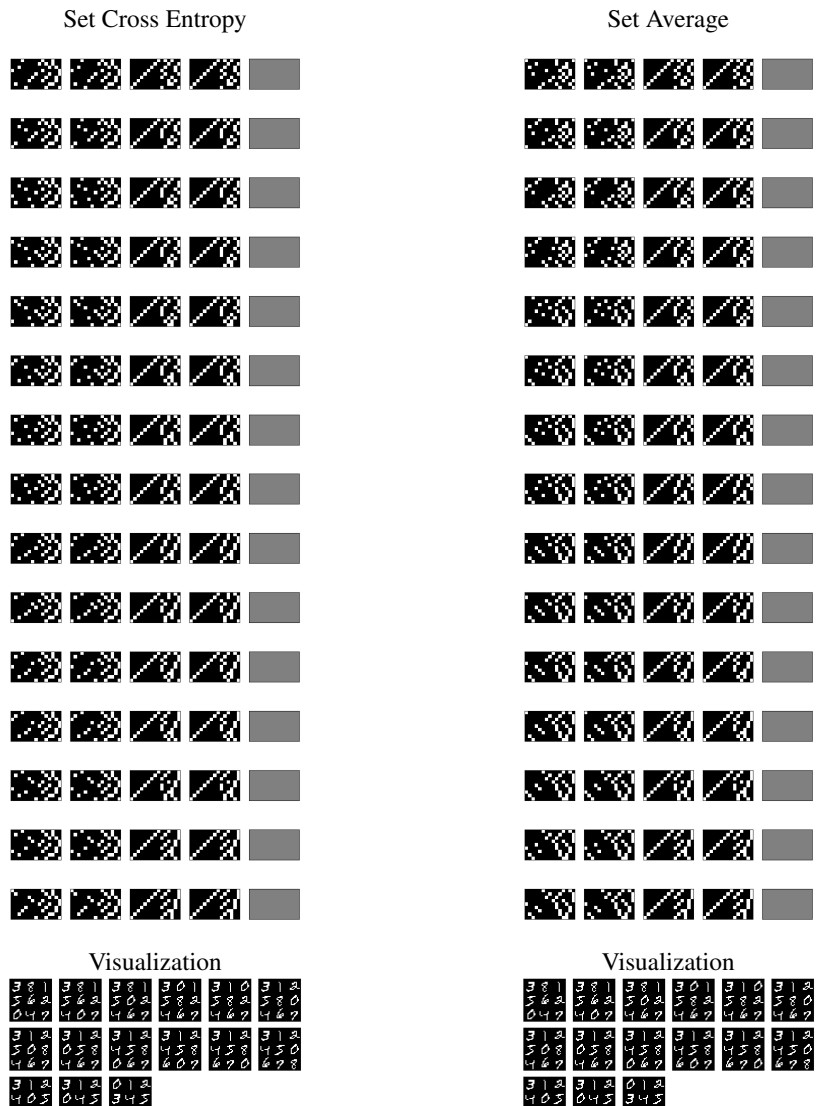

Problem 001, generated by 014 steps

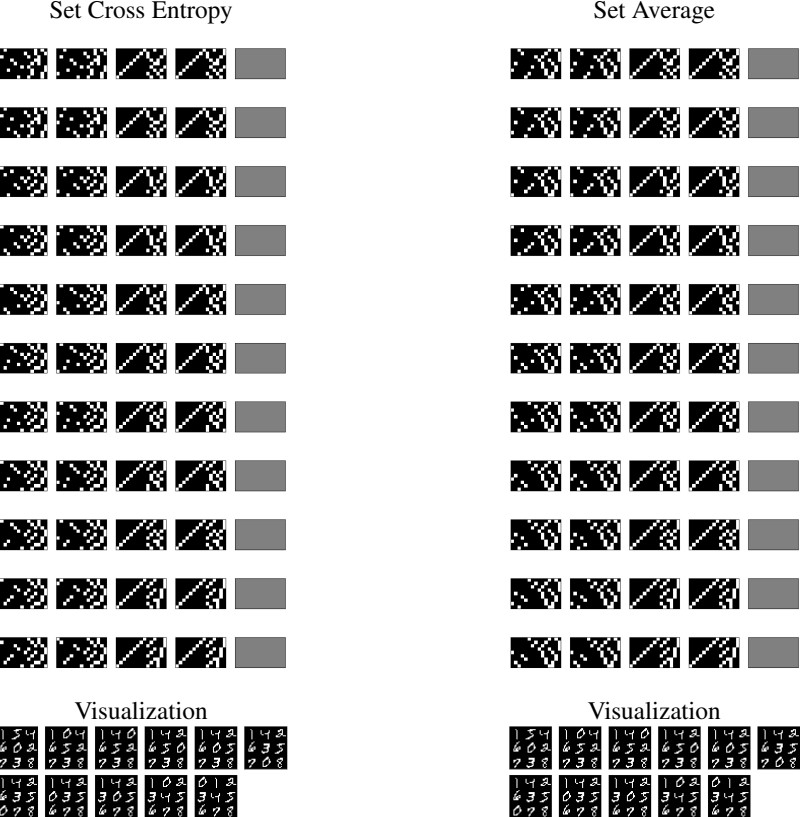

Problem 002, generated by 014 steps

Set Cross Entropy                    Set Average

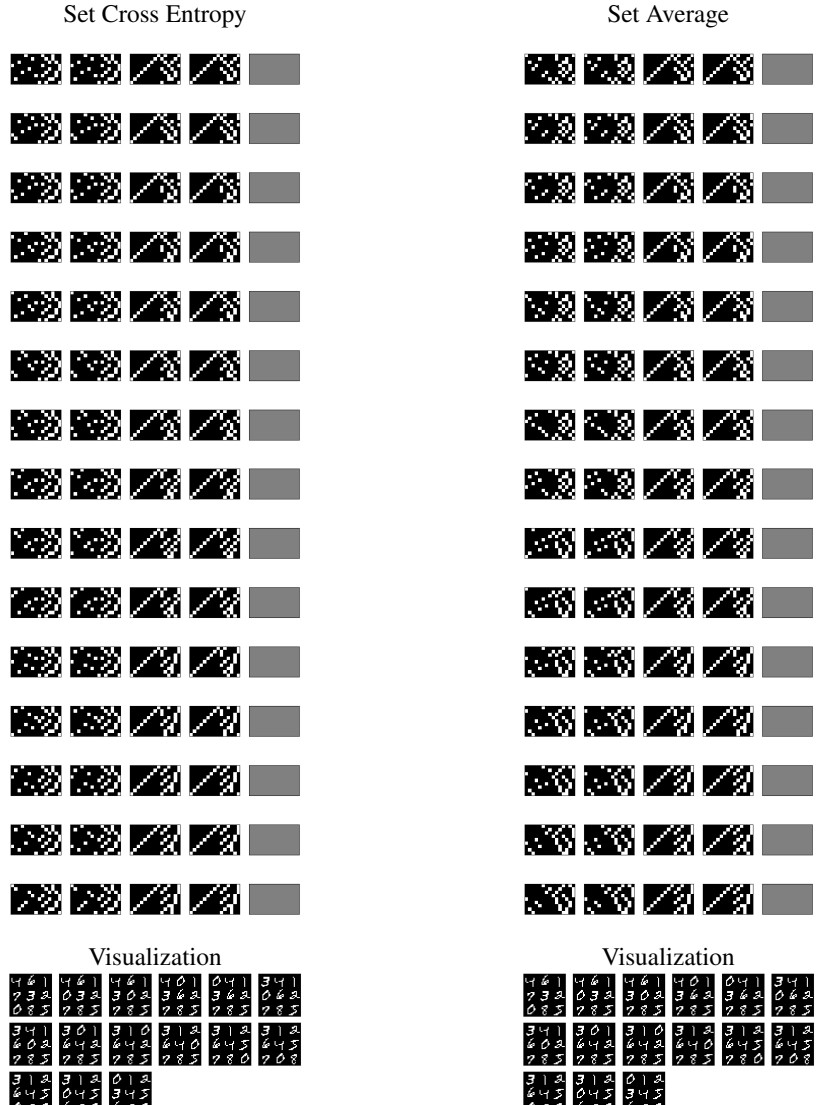

Visualization                    Visualization

Problem 003, generated by 014 steps

Set Cross Entropy                    Set Average

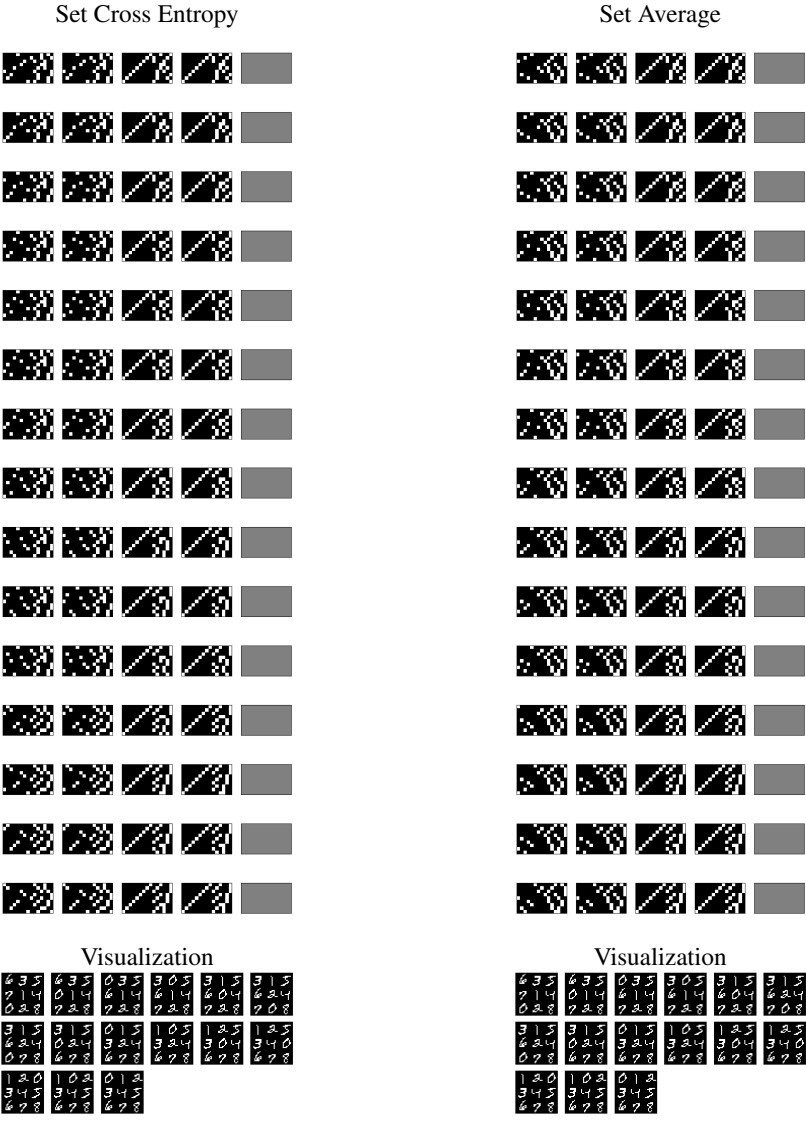

Visualization                         Visualization

Problem 004, generated by 014 steps

Set Cross Entropy                                    Set Average

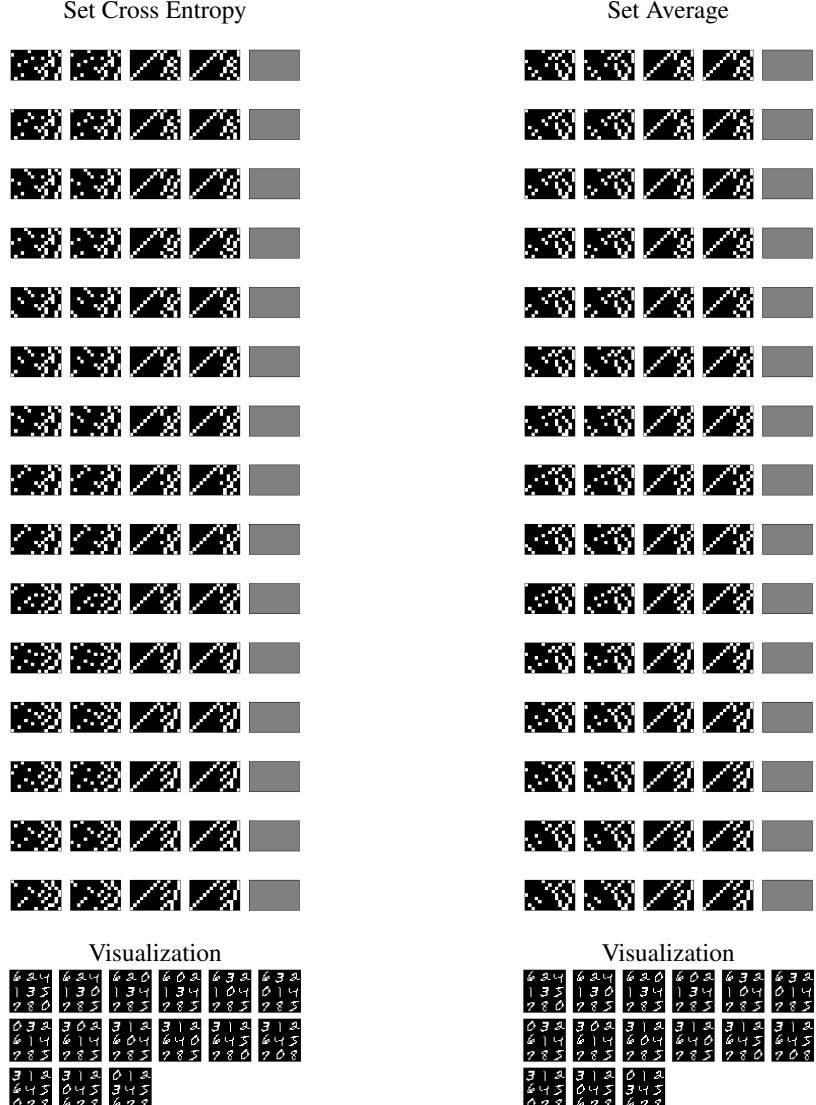

Visualization                                        Visualization

Problem 005, generated by 014 steps

Set Cross Entropy            Set Average

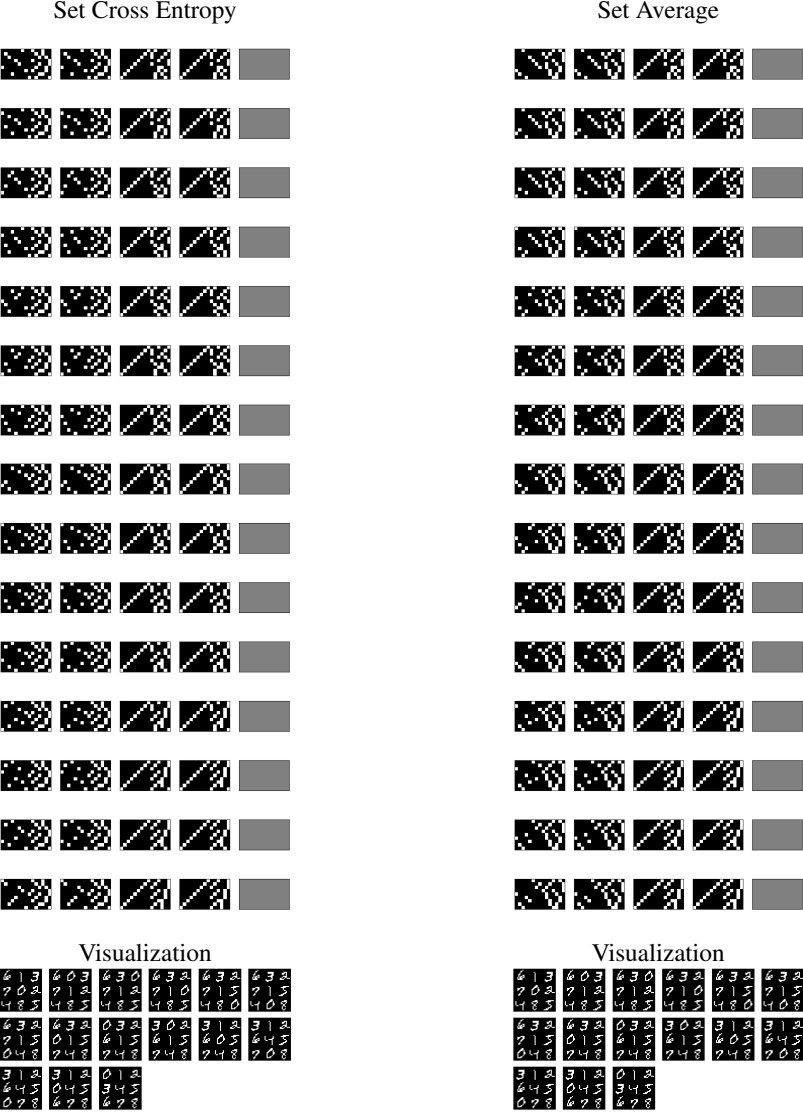

Visualization            Visualization

Problem 006, generated by 014 steps

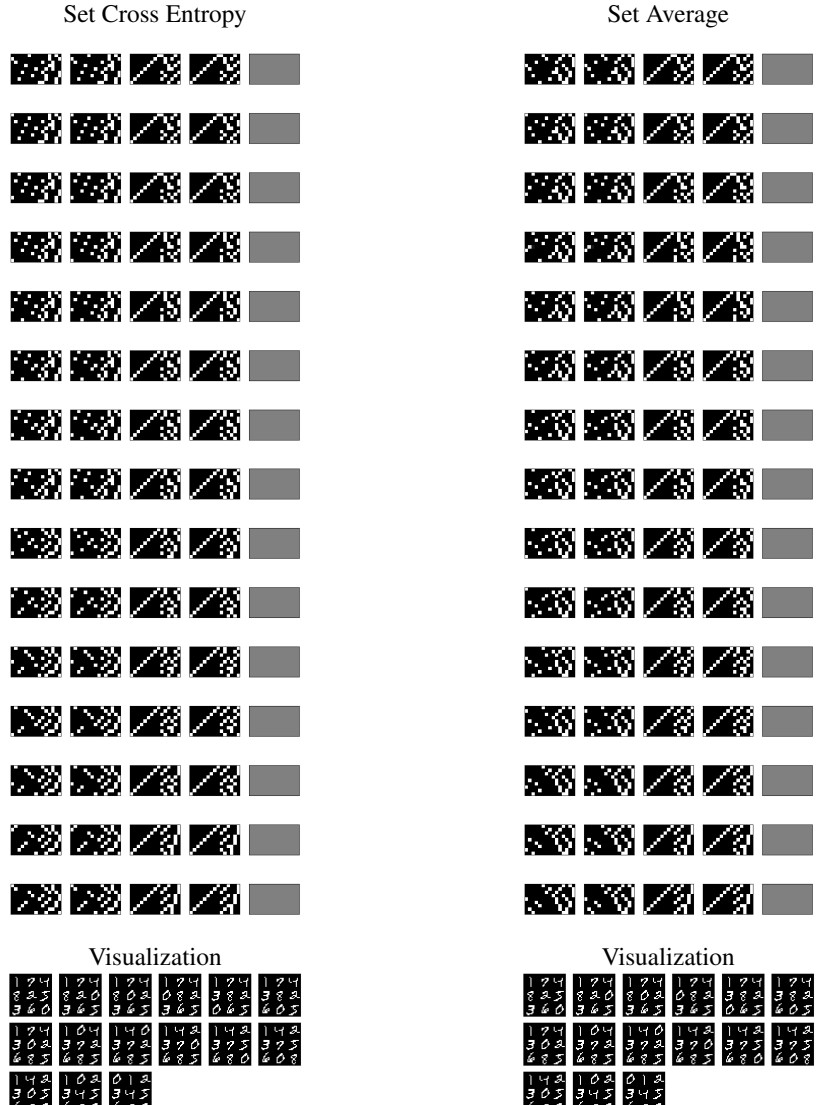

Set Cross Entropy

Set Average

Visualization

Visualization

Problem 007, generated by 014 steps

Set Cross Entropy                                    Set Average

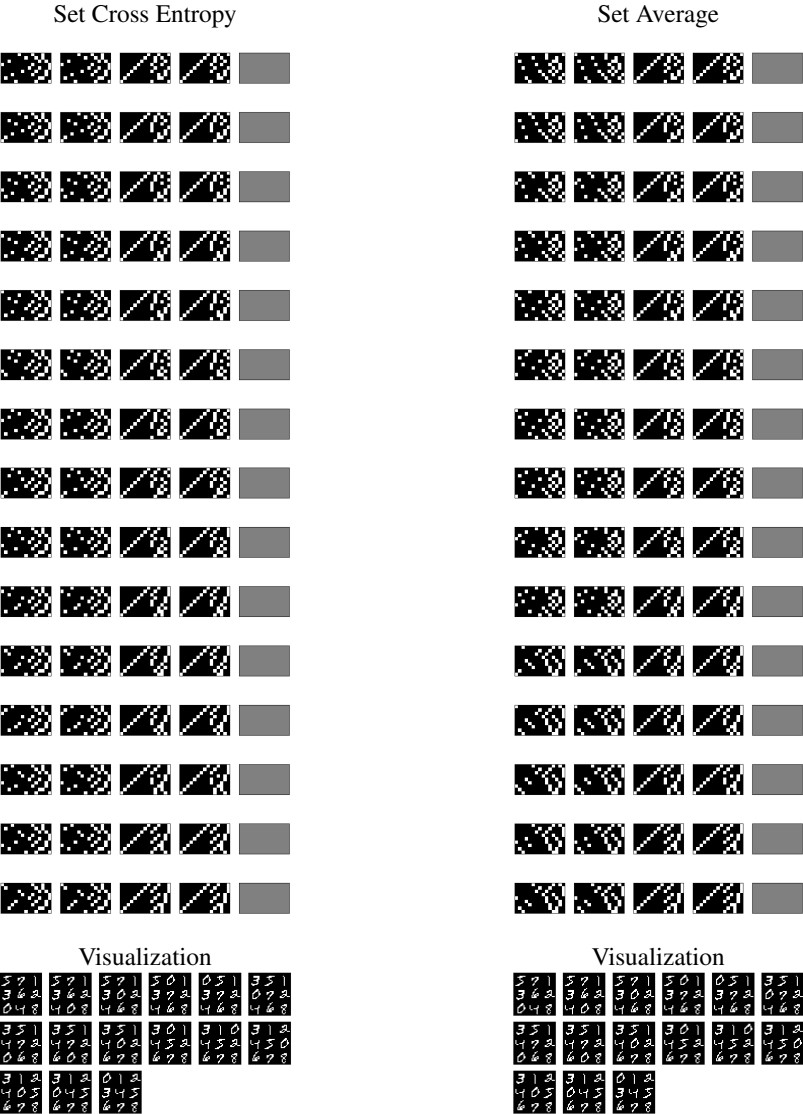

Visualization                                        Visualization

Problem 008, generated by 014 steps

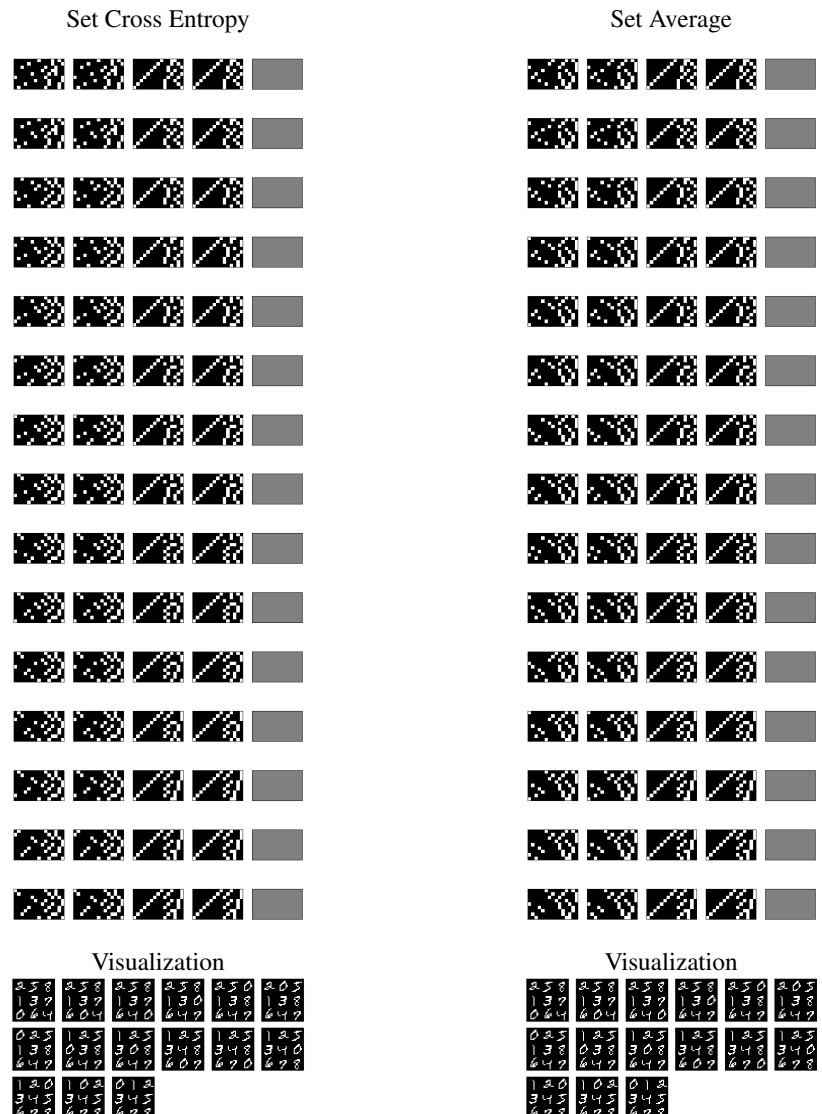

Problem 009, generated by 014 steps

Set Cross Entropy                                    Set Average

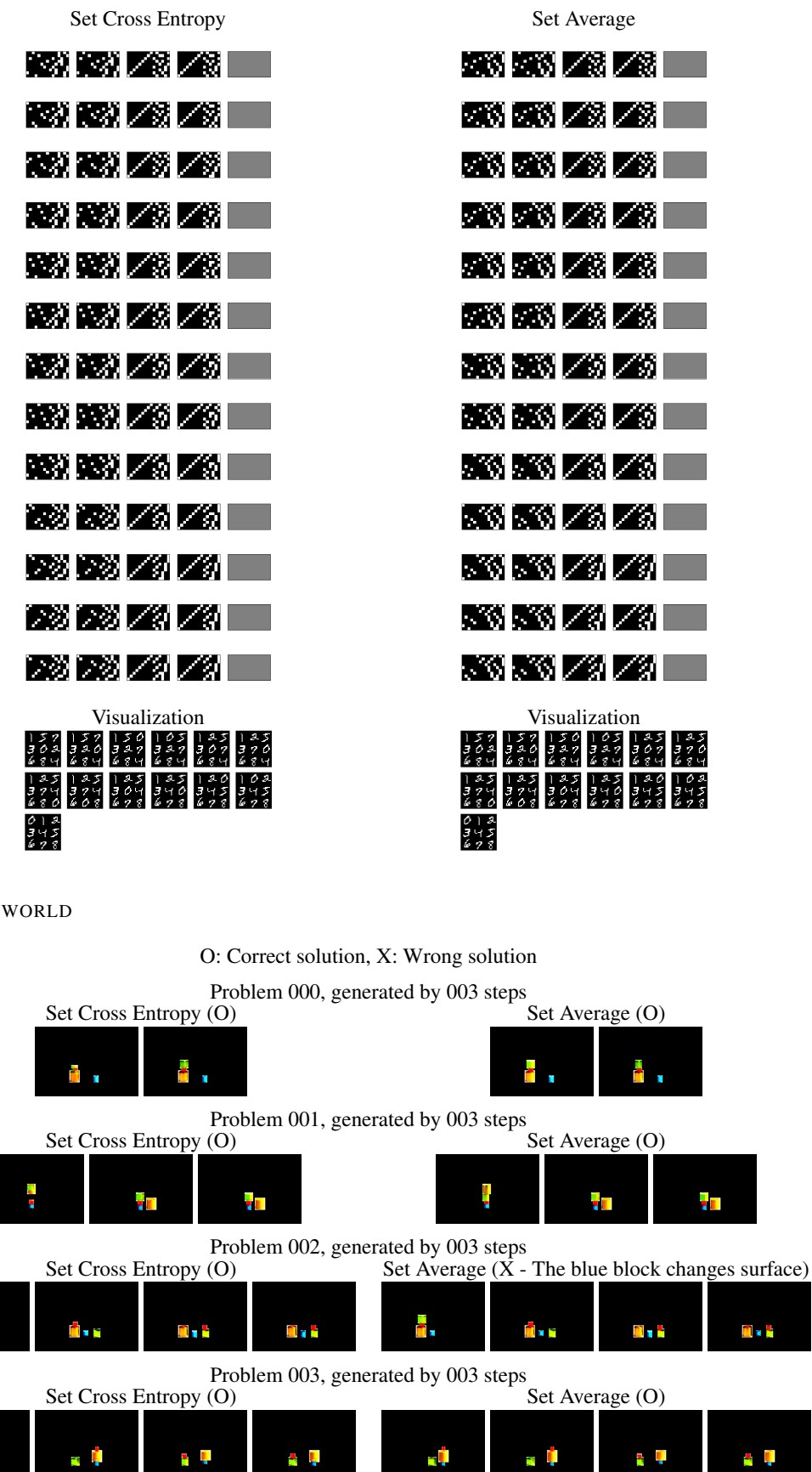

Visualization                                        Visualization

BLOCKSWORLD

O: Correct solution, X: Wrong solution

Problem 000, generated by 003 steps
Set Cross Entropy (O)                                Set Average (O)

Problem 001, generated by 003 steps
Set Cross Entropy (O)                                Set Average (O)

Problem 002, generated by 003 steps
Set Cross Entropy (O)                Set Average (X - The blue block changes surface)

Problem 003, generated by 003 steps
Set Cross Entropy (O)                                Set Average (O)

Problem 004, generated by 003 steps

Set Cross Entropy (X)                    Set Average (X)

Problem 005, generated by 003 steps

Set Cross Entropy (X)                    Set Average (O)

Problem 006, generated by 003 steps

Set Cross Entropy (O)                    Set Average (O)

Problem 007, generated by 003 steps

Set Cross Entropy (O)                    Set Average (O)

Problem 008, generated by 003 steps

Set Cross Entropy (O)                    Set Average (O)

Problem 009, generated by 003 steps

Set Cross Entropy (O)                    Set Average (O)

Problem 000, generated by 007 steps

Set Cross Entropy (O)                    Set Average (X)

Problem 001, generated by 007 steps

Set Cross Entropy (O)                    Set Average (O)

Problem 002, generated by 007 steps

Set Cross Entropy (X)                    Set Average (X)

Problem 003, generated by 007 steps

Set Cross Entropy (X)                    Set Average (X)

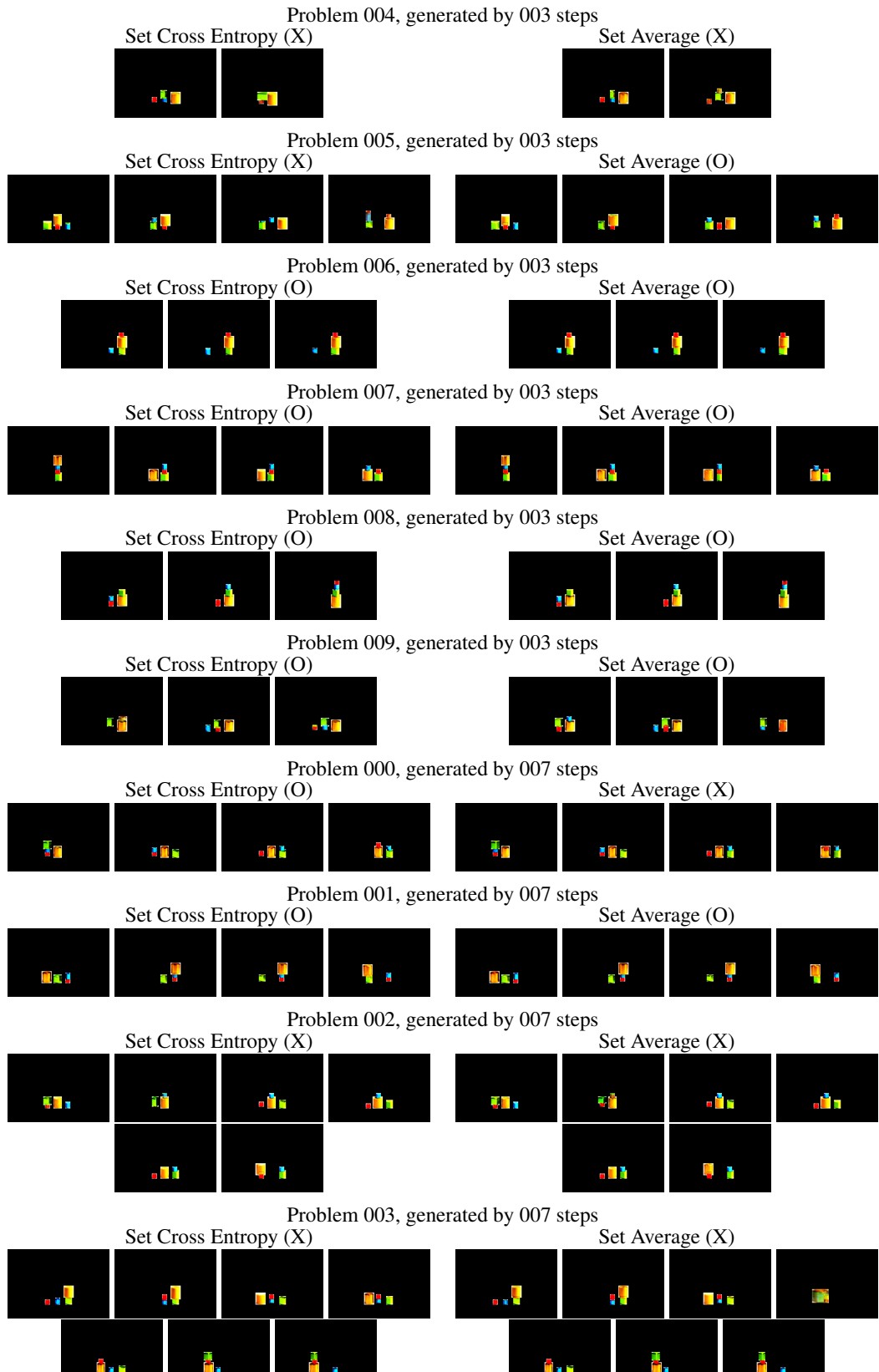

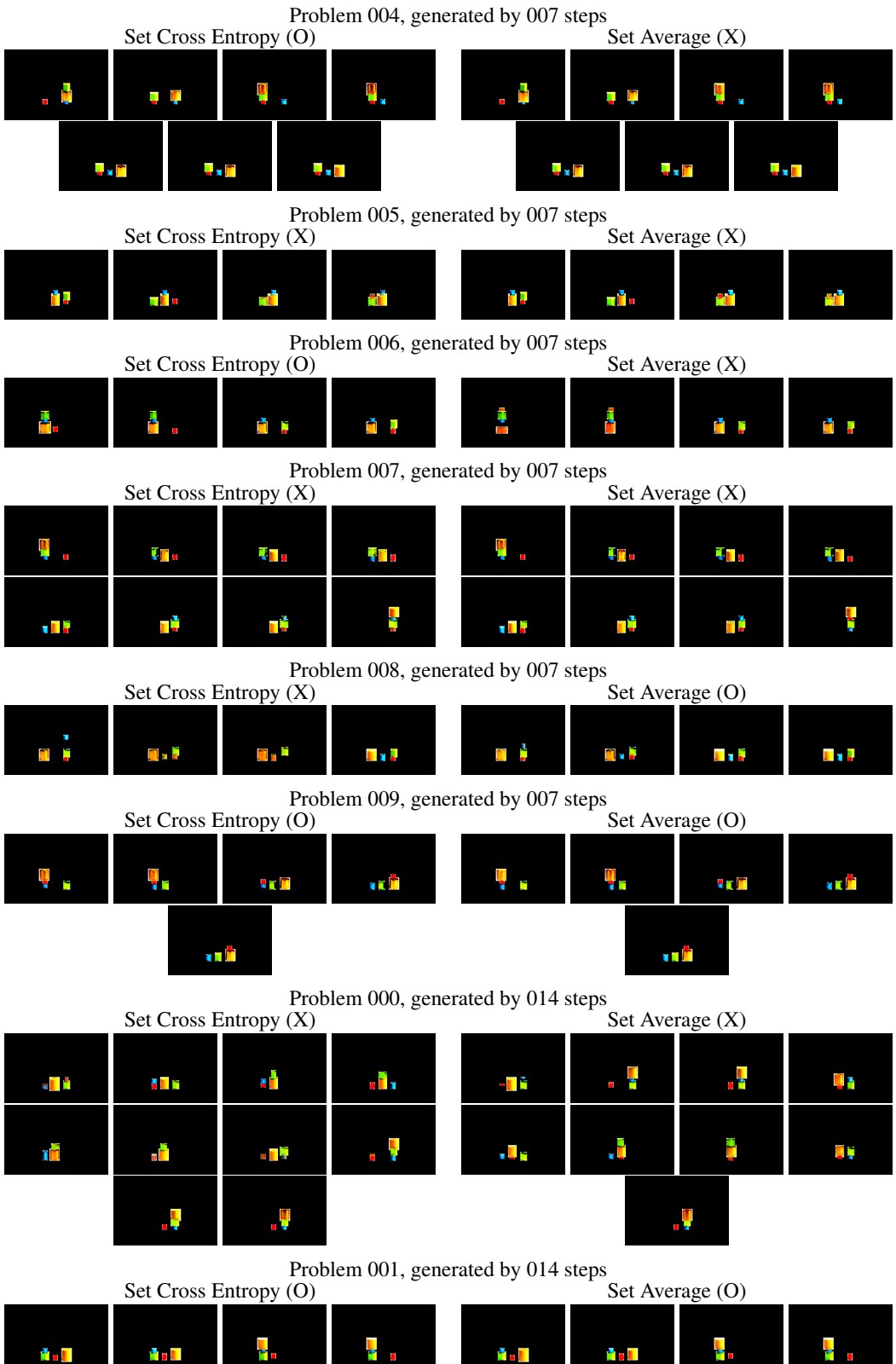

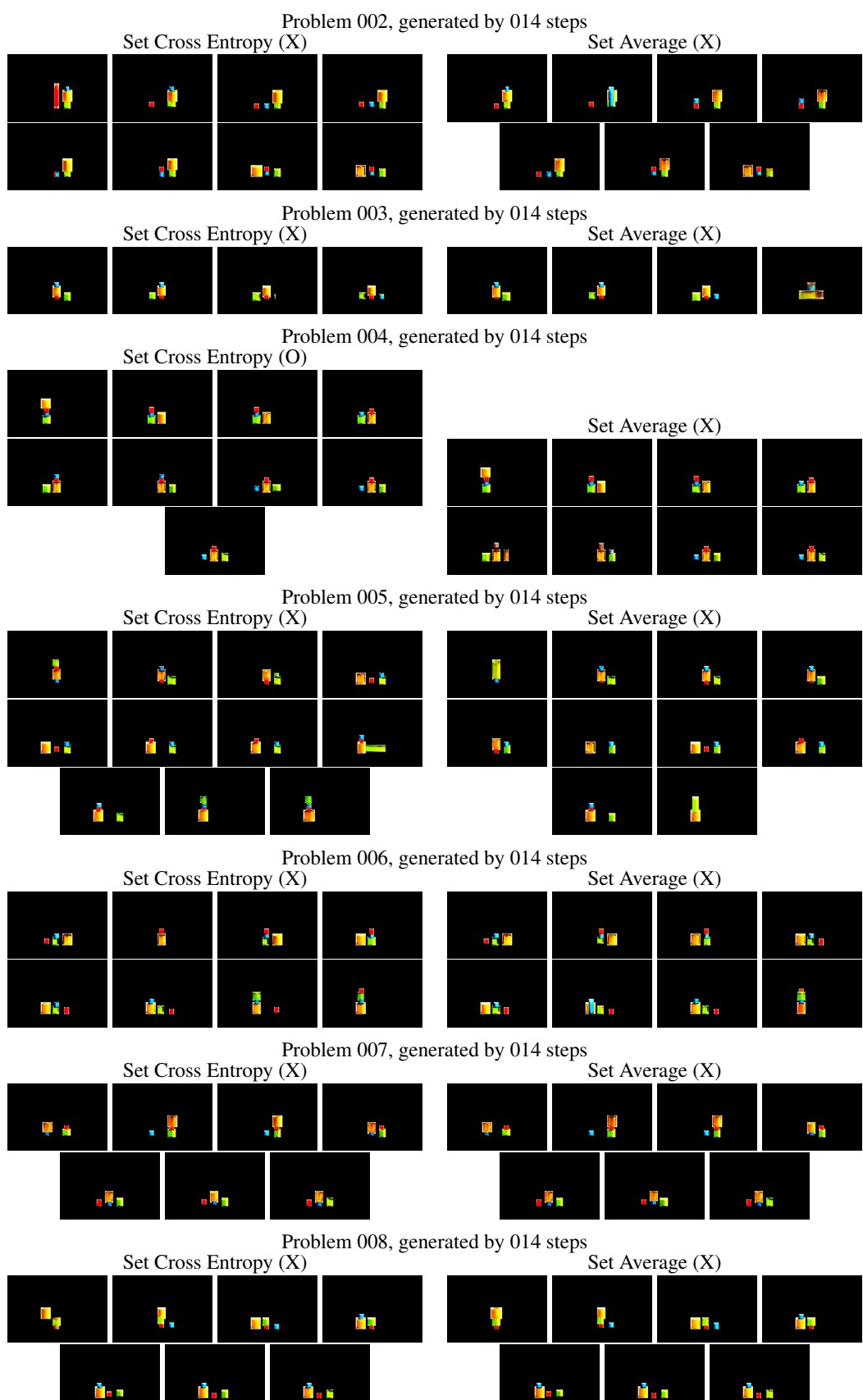

Problem 009, generated by 014 steps

Set Cross Entropy (X)                    Set Average (X)

