# OpenReview forum: "Likelihood-based Permutation Invariant Loss Function for Probability Distributions"
_ICLR.cc/2019/Conference_

### Official Review · AnonReviewer2 · 2018-10-23
**Lacks clarity**

**Rating:** 4
**Confidence:** 4

**Review:**

In the manuscript entitled "Likelihood-based Permutation Invariant Loss Function for Probability Distributions" the authors propose a loss function for training against instances in which ordering within the data vector is unimportant.  I do not find the proposed loss function to be well motivated, find a number of confusing points (errors?) in the manuscript, and do not easily follow what was done in the examples.

First, it should be noted that this is a very restricted consideration of what it means to compare two sets since only sets of equal size are under consideration; this is fundamentally different to the ambitions of e.g. the Hausdorff measure as used in analysis.  The logsumexp formulation of the proposed measure is unsatisfactory to me as it directly averages over each of the independent probabilities that a given element is a member of the target set, rather than integrating over the combinatorial set of probabilities for each set of complete possible matches.  Moreover, the loss function H() is not necessarily representative of a generative distribution.

The definition of the Hausdorff distance given is directional and is therefore not a metric, contrary to what is stated on page 2.

I find the description of the problem domain confusing on page 3: the space [0,1]^NxF is described as binary, but then values of log y_i and log (1-y_i) are computed with y in [0,1] so we must imagine these are in fact elements in the open set of reals: (0,1).

Clarity of the examples could be greatly improved, in particular by explaining precisely what is the objective of each task and what are the 'ingredients' we begin with.

---

> ### Author Response · Authors · 2018-11-12
> **reply**
>
>
>   > only sets of equal size are under consideration ...
>
>   See the overall reply. It is not only for sets of equal size.
>
>
>   > The logsumexp formulation of the proposed measure is unsatisfactory
>   > to me as it directly averages over each of the independent
>   > probabilities that a given element is a member of the target set,
>   > rather than integrating over the combinatorial set of probabilities
>   > for each set of complete possible matches.
>
>   We have shown (the proof in page.3, sec.3, par.7) that our method
>   guarantees that, at the global minima, every element x of the dataset
>   X is matched by some element y of the output Y exactly once, just as
>   in Hausdorff measure.
>
>   We would like to hear more details about why it is unsatisfactory despite the guarantee.
>
>   > The definition of the Hausdorff distance given is directional and is
>   > therefore not a metric, contrary to what is stated on page 2.
>
>   We already clarified this in page 2: "Note that, in this work, we use
>   the informal usage of the terms “distance” or “metric” ..."  We
>   also only stated that Hausdorff distance is a metric, and not that the
>   directed Hausdorff distance is a metric.  We moved the clarification
>   to the beginning of the section to avoid confusion.
>
>   > the space [0,1]^NxF is described as binary...
>
>   See the overall reply. [0,1] is a closed set of reals, not discrete
>   values.  The input is not assumed to be discrete. "binomial
>   distribution" might be the more appropriate term. We rephrased them in the revision.
>
>   > what is the objective of each task and what are the 'ingredients' we
>     begin with.
>
>   The detailed descriptions are in the appendix.  The first two tasks (8
>   puzzle, Blocksworld) are the autoencoding task, but without
>   considering the ordering in the first axis of the data point (i.e. the order of the elements).
>   In 8-puzzle, the object representation (an element of the set) is
>   hand-crafted as in Fig.1.  In Blocksworld, a set of objects are
>   extracted from the image, and the image patch and the bounding box
>   information are compressed into 1224-D vector by a feature engineering
>   using Conv-AE (Appendix, 6.3).
>
>   The last task (rule learning task) is to predict a set of terms from a
>   single term, where each term is a n-hot vector representing a
>   first-order logic term (Section 4.2, Appendix 6.4).  We moved the
>   description in the appendix to the main text.

---

> > ### Comment · AnonReviewer2 · 2018-11-17
> > **some improvements to clarity**
> >
> > The authors have clarified a number of the confusing points; however, I'm still not satisfied with the objective: the representation of the logsumexp that comes after the proof on page 3, namely the manipulations done in equation (4).

---

> > > ### Author Response · Authors · 2018-11-19
> > > **reply**
> > >
> > >   Thank you for the further comments.
> > >
> > >   Getting back to the first comment,
> > >
> > >   > The logsumexp formulation of the proposed measure is unsatisfactory
> > >   > to me as it directly averages over each of the independent
> > >   > probabilities that a given element is a member of the target set,
> > >   > rather than integrating over the combinatorial set of probabilities
> > >   > for each set of complete possible matches.
> > >
> > >   Since every sets have the equal number of permutations (assuming they
> > >   are of the same size and every elements are distinct), the individual
> > >   probability and such an integrated sum over every permutations of the
> > >   probabilities will only differ by a constant factor scaling.
> > >
> > >   For example, P([x,y,z]=[1,2,3]) vs sum of all permutations,
> > >   P([x,y,z]=[1,2,3])+…+P([x,y,z]=[3,2,1]) differs by 6, the number of
> > >   permutations for 3 elements.  Since other patterns, such as
> > >   P([x,y,z]=[4,5,6]), also have 6 permutations, considering just one
> > >   ordering and considering all ordering makes essentially no difference.
> > >
> > >   Informally speaking, the key to understand our method is to notice
> > >   that considering the exponential/combinatorial number of possible
> > >   permutations (matches) is equivalent to treating every permutation as
> > >   a distinct event, which is not necessary when the output order is
> > >   ignored.  It is not something that is asked for in this
> > >   problem setting.

---

### Official Review · AnonReviewer1 · 2018-10-23
**Extension to case of sets with different sample sizes?**

**Rating:** 6
**Confidence:** 3

**Review:**

The paper is understandable and the question addressed is interesting. The use of log likelihoods to metrize distances between sets, although not new, is used quite effectively to address the issue of label switching in sets. Although the run time is O(N^2), the metric can be computed in a parallelized manner. The question of comparing sets of different sample sizes would be a valuable extension to the work. Although I think the proposed loss function addresses some important issues, would like to defer the question of acceptance/rejection to other reviewers due to lack of expertise in related areas.

---

> ### Author Response · Authors · 2018-11-12
> **Overall comment**
>
>   > The use of log likelihoods to metrize distances between sets,
>     although not new, ...
>
>   Log likelihood itself is a general notion and is not new.  However,
>   our contribution is formulating the log likelihood for sets, with a
>   proof that it converges to the correct answer at the global minima.
>
>   > The question of comparing sets of different sample sizes would be a
>   > valuable extension to the work.
>
>   See the overall reply. We added some explanations and also a
>   new experiment that shows that it can model the sets with the different number of elements.

---

### Official Review · AnonReviewer3 · 2018-10-24
**new loss function for set autoencoders; experiments are not sufficient**

**Rating:** 5
**Confidence:** 4

**Review:**

This paper proposes an objective function for sets autoencoders such that the loss is permutation invariant with respect to the order of reconstructed inputs. I think that the problem of autoencoding sets is important and designing custom loss functions is a good way to approach it. Thus, I quite like the idea of SCE  from that point of view. However, I find the experiments not convincing for me to accept the paper.

While reading Section 3, I found it hard to keep in mind that x and y are discrete probability distributions and the notation like P(x=y) is not making things easier. Actually, I’ve never seen cross entropy written with P(x=y). Though is my personal opinion and I don’t have a suggestion on how to improve the explanations in Eq. 1-8. However, I’m glad there is an example at the end of Section 3.

I have some comments on the Experiments section.

* Puzzles:
(1) Figure 1 could have been prettier.
(2) The phrase “The purpose of this experiment is to reproduce the results from (Zaheer et al., 2017)” makes little sense to me.  In Deep Sets, there are many experiments and it’s not clear which experiment is meant here.
(3) Table 1 gives test error statistics for 10 runs. What is changed in every run? Does the test set stay the same in every run or is a kind of a cross-validation? Or is it just a different random seed for the initial weights? I could not find an explanation in the text, so there is no way I can interpret the results.

* Blocksworld: the reconstructions are nice, but the numbers in Table 2 are difficult to interpret.
For example, I cannot estimate how important the difference of 10 points in SH scores is.

* Rule learning ILP tasks: I don’t know enough about learning logic rules tasks to comment on those experiments, but Table 3 seems overwhelming and the concept of 10 runs is still unclear.

--- General comment on the experiments ---

I think an important goal of any autoencoder is to learn a representation that can be useful in other tasks. There is even an example in the paper: “set representation of the environment is crucial in the robotic systems”. Thus, the experiments I would like to see are about evaluating the quality of a representation from an SCE-trained autoencoder compared to other training methods.  Without those experiments, I cannot estimate how valuable the SCE loss function is.

---

> ### Author Response · Authors · 2018-11-12
> **Overall reply**
>
>   > Actually, I’ve never seen cross entropy written with P(x=y).
>
>   The usual notation of the probability distribution, such as P(x), is
>   an abbreviation of a random variable X taking a particular value x,
>   i.e. P(X=x), following the notation in Ian Goodfellow's Deep Learning
>   textbook, chap.3
>   ([http://www.deeplearningbook.org/contents/prob.html]). We just kept
>   it expanded for denoting the cross entropy.
>
>   > The phrase “The purpose of this experiment is to reproduce the
>   > results from (Zaheer et al., 2017)” makes little sense to me.
>   > In Deep Sets, there are many experiments and it’s not clear which
>   > experiment is meant here.
>
>   By "reproduce" we did not mean we run the same experiment in the prior
>   work; What we did is that we confirmed their general claim (their
>   particular network structure is able to encode an input in a
>   permutation invariant manner) in *our* experimental setting.
>
>   We rephrased it in the revision in order to avoid the confusion.
>
>   > Table 1 gives test error statistics for 10 runs. What is changed in
>     every run?
>
>   The purpose of running the experiments 10 times is to address the
>   potential concern about the stability of the training.
>
>   We kept the same set of training/testing data, the only difference is
>   the random seed.  In one of the 10 runs, A1H (set average pseudo
>   metric) did not converge, showing that the A1H (baseline) could be
>   unstable, possibly due to the issue explained in the example at the
>   end of section 3. # , though this is a speculation from the empirical
>   result.  This shows another empirical evidence that our proposed
>   method is superior.  We clarified these points in the revision.
>
>   > Blocksworld: the reconstructions are nice, but the numbers in Table
>     2 are difficult to interpret.
>
>   We agree with this point. To address it, we added the RMSE between the
>   visualized pictures to give more insights.  Note that these pictures
>   are not the direct output of the neural network; However, comparing
>   the reconstructed pictures by RMSE should give some intuitive sense
>   since the error directly translates to the pixel value.  A new table
>   is added in the revision.
>
>   Furthermore, we compared the visualized results between the networks
>   trained with a different loss formulation.
>
>   Since we believe the same issue applies to 8-puzzles, we also added a
>   new evaluation metric for 8-puzzles: Since we know 8 puzzle feature
>   vectors are discrete (a domain knowledge, not the assumption in our
>   proposed method), we can directly compare the output reconstruction
>   with the input by rounding the continuous output to 0/1 and comparing
>   whether all elements are correctly reconstructed, and count the rate
>   of the successful reconstructions across the dataset.  Another new
>   table is added in the revision.
>
>   > Rule learning ILP tasks: the concept of 10 runs is still unclear.
>
>   This is same as the previous experiments; The only difference is the
>   random seed.
>
>   > I think an important goal of any autoencoder is to learn a
>   > representation that can be useful in other tasks. There is even an
>   > example in the paper: “set representation of the environment is
>   > crucial in the robotic systems”. Thus, the experiments I would like
>   > to see are about evaluating the quality of a representation from an
>   > SCE-trained autoencoder compared to other training methods.
>   > Without those experiments, I cannot estimate how valuable the SCE
>   > loss function is.
>
>   We were surprised by this question.
>
>   First, we do not focus only on the autoencoding task. In the ILP task,
>   the neural network learns to predict a set from the single
>   element. For example, in the last `neighbor5` experiment in Table 6,
>   the task is to predict 5 elements from 1 element.  Our contribution is
>   the method for training a NN to output a set, not limited
>   to autoencoding.
>
>   Regarding the value of reconstructing a set, existing work (Vinyal
>   NIPS 2015, ICLR 2016) already showed that once we can train a NN to
>   output a set (by whatever means), it allows a variety of tasks to be
>   solved.  The problem is that existing methods relied on an ad-hoc
>   preprocessing and/or a careful tuning that depends on the domain
>   knowledge, or a sequential process such as Gale-Shapley.
>   Our contribution is to completely remove these assumptions motivated
>   by the theoretical formulation, NOT by an empirical success
>   that may occur by chance in a particualr problem setting.

---

> > ### Comment · AnonReviewer3 · 2018-11-22
> > **reply to rebuttal**
> >
> > Thank you for the rebuttal! It clarified some things, but there are still parts I'm very confused about.
> >
> > >> The usual notation of the probability distribution, such as P(x), is an abbreviation of a random variable X taking a particular value x, i.e. P(X=x)
> >
> > I agree with that, but the confusion is coming from the fact that here x is a random variable and x also denotes a probability distribution starting from Eq. 1 if I'm not mistaken.
> >
> > >> The input is not assumed to be discrete. "binomial distribution" might be the more appropriate term. We rephrased them in the revision.
> >
> > I don't see where the binomial distribution comes from.
> >
> > >> Regarding the value of reconstructing a set, existing work (Vinyal NIPS 2015, ICLR 2016) already showed that once we can train a NN to output a set (by whatever means), it allows a variety of tasks to be
> >   solved.
> >
> > I don't know about pointer nets, but I think models like seq2seq for sets struggle more with encoding a set in a permutation-invariant manner. I still think that evaluating the quality of representations learnt with an SCE loss is not a very surprising thing to ask for.
> >
> > I'm sure that SCE loss function is a great idea, but I agree with Reviewer2 about the clarity. So I'd keep my score for now.

---

> > > ### Author Response · Authors · 2018-11-26
> > > **Planning with Plannable Representation, i.e., Discrete Latent Binary Representation + Latplan system (AAAI18)**
> > >
> > >   > I still think that evaluating the quality of representations learnt
> > >     with an SCE loss is not a very surprising thing to ask for.
> > >
> > >   To address the reviewer's request, we are running a new experiment. We
> > >   modified Latplan (Asai, Fukunaga AAAI18) neural-symbolic classical
> > >   planning system, a system that operates on a discrete symbolic latent
> > >   space of the real-valued inputs and runs dijkstra/A* search using a
> > >   state-of-the-art symbolic classical planning solver. We modified
> > >   Latplan to take the set-of-object-feature-vector input rather than
> > >   images.  It is a high-level task planner (unlike motion planning /
> > >   actuator control) that has implications on robotic systems.
> > >
> > >   To briefly describe the Latplan system, it learns the discrete binary
> > >   latent space of an arbitrary raw input (e.g. images) with a
> > >   Gumbel-Softmax variational autoencoder, learns a discrete transition
> > >   model in the state space from the transition examples, and runs a
> > >   systematic, complete search algorithm such as Dijkstra search or A*
> > >   which gurantee the optimality of the solution.  Unlike RL-based
> > >   planning systems, the search agent does not contain the learning
> > >   aspects.  The discrete plan in the latent space is mapped back to the
> > >   raw image visualization of the plan execution.  A similar system
> > >   replacing Gumbel Softmax VAE with Causal InfoGAN was later proposed
> > >   (Kurutach et. al., NIPS18).
> > >
> > >   We replaced Latplan's Gumbel-Softmax VAE with our autoencoder used in
> > >   the 8-Puzzle and the Blocksworld experiments (Appendix, Sec 6.1). Our
> > >   autoencoder also uses Gumbel Softmax in the latent layer, but it uses
> > >   (Zaheer 2017) encoder and is trained with Set Cross Entropy in order
> > >   to encode permutation-invariant information.
> > >
> > >   When the network learned the representation, it guarantees that the
> > >   planner finds a solution because the search algorithm being used
> > >   (e.g. Dijkstra) is a complete, systematic, symbolic search algorithm,
> > >   which guarantees to find a solution whenever it is reachable in the
> > >   state space.  If the network cannot learn the permutation-invariant
> > >   representation, the system cannot solve the problem and/or return the
> > >   human-comprehensive visualization. This makes the specific
> > >   permutation-invariant representation using (Zaheer 2017) and the
> > >   proposed Set Cross Entropy necessary when the input is given as a set
> > >   of feature vectors.
> > >
> > >   The new results will be found in the appendix section of the revision.
> > >   The reason for putting it in the appendix is that the value of the new
> > >   experiment added to our main contribution could be debatable.
> > >   Our MAIN claim is a loss function that can train a NN to reconstruct a
> > >   set, and the importance of better handling a set in the context of
> > >   deep learning is already widely recognized in the ML community (see
> > >   the existing work cited by our paper).

---

> > > ### Author Response · Authors · 2018-11-26
> > > **Paper updated**
> > >
> > > Please check the updated paper for the additional results.

---

### Author Response · Authors · 2018-11-12
**Overall reply to all reviewers**

Thanks very much for the thoughtful comments.  We first discuss the
general response to the reviewers, then in the threads we reply to the
specific concerns raised by each reviewer.

While ICLR may not mandate it, we would appreciate it if you confirm in
the reply that you read all of the rebuttals, including the replies to
the other reviewers.

To Reviewer #1 and #3:

The input is *not* assumed to be discrete.  [0,1] is meant to be a
closed set of reals between 0 and 1 (while an open set is denoted as
(0,1)).  We clarified this in the revision.  By "binary representation"
in page 3, we meant a binomial probability distribution (thus can be a
continuous value) in contrast to the multinomial
probability distribution.

In fact, each input feature vector in the Blocksworld experiment are
generated from the individual image patches by an additional autoencoder
with a sigmoid latent activation (Appendix 6.3, Figure 4), thus they are
in fact the continuous vectors.

To Reviewer #2 and #3:

> #2 The question of comparing sets of different sample sizes would be a
> valuable extension to the work.

> #3 only sets of equal size are under consideration ...

(Sec 3, paragraph 3) When the size of the set varies, we can add an
arbitrary number of artificially generated distinct dummy elements to
the set in order to keep the size of the set equal throughout
the dataset.

For example, when there is a set of N objects of F features and we want
to normalize the size of the set to N' (>N), one way is to add an
additional axis to the feature vector (F+1 features) where the
additional F+1-th feature is 0 for the real data and 1 for the dummy
data, and the distinct N'-N objects are generated in an arbitrary way
(e.g. as a binary sequence 100000, 100001, 100010, 100011, ... for F=5).
During the inference, the dummy vectors in the output can be removed.

We added this explanation in the revision.

The ability to learn from such an augmented dataset is only a matter of
the neural network topology and the hyperparameter tuning, which is not
the main topic of this paper.  Whether using an LSTM, a CNN or a fully
connected network does not affect the loss formulation presented in this
work.  Given enough capability for tuning and network engineering (which
includes the choise of NN), they should also be able to learn from such
an augmented dataset, as the augmented dataset has the
same characteristics.

However, to address this concern, we ran an additional experiment where
some of the tiles are randomly missing in 8 puzzles.  The feature
vectors are extended from 15 to 16 dimensions and the dummy elements are
inserted as explained above.  The system reconstructs the elements
including the dummy ones, which means it can identify how many and which
elements are missing for the given augmented output. Moreover, the
proposed Set Cross Entropy outperformed Set Average and other loss
functions.  See Section 4.1 for the additional details and the results.

Finally, we noticed that the evaluation metric for the
ILP task was not explained properly; Originally, we wrote:

> We counted the ratio of the clauses across the test set where every
  output term matches against one of the body terms.

This is wrong: It is an improper way to measure the correct matching, as
there could be duplicates.  Our implementation did not measured the
success rate in this way.  The correct description that reflects our
implementation is:

> We counted the ratio of the clauses across the test set where every
  *body* term matches against one of the *output* terms.

---

### Meta-Review · Area_Chair1 · 2018-12-13
**Borderline paper**

**Confidence:** 3
**Recommendation:** Reject

**Metareview:**

This paper proposes a new permutation invariant loss (where the order doesn't matter), motivated by set autoencoding settings. This is an important problem, and the authors' solution is interesting.  The reviewers, however, found the exposition to be unclear, in particular the explanation on how the loss function is derived was confusing for two of the reviewers. Reviewers also found the experimental results to be not convincing, even after the revision. This is a borderline paper: the idea is valuable and I'd encourage the authors to develop it further, improving exposition and including additional experiments as suggested by the reviewers.